# Functional impact and molecular binding modes of drugs that target the PI3K isoform p110δ

Floyd Hassenrück [1,2,3,5], Maria Farina-Morillas [1,2,3], Lars Neumann[1,2,3], Francesco Landini[1,2,3], Stuart James Blakemore[1,2,3], Mina Rabipour[1,2,3], Juan Raul Alvarez-Idaboy [4], Christian P. Pallasch[1,2,3], Michael Hallek [1,2,3], Rocio Rebollido-Rios [1,2,3,5✉] & Günter Krause [1,2,3,5✉]

Targeting the PI3K isoform p110δ against B cell malignancies is at the mainstay of PI3K inhibitor (PI3Ki) development. Therefore, we generated isogenic cell lines, which express wild type or mutant p110δ, for assessing the potency, isoform-selectivity and molecular interactions of various PI3Ki chemotypes. The affinity pocket mutation I777M maintains p110δ activity in the presence of idelalisib, as indicated by intracellular AKT phosphorylation, and rescues cell functions such as p110δ-dependent cell viability. Resistance owing to this substitution consistently affects the potency of p110δ-selective in contrast to most multi-targeted PI3Ki, thus distinguishing usually propeller-shaped and typically flat molecules. Accordingly, molecular dynamics simulations indicate that the I777M substitution disturbs conformational flexibility in the specificity or affinity pockets of p110δ that is necessary for binding idelalisib or ZSTK474, but not copanlisib. In summary, cell-based and molecular exploration provide comparative characterization of currently developed PI3Ki and structural insights for future PI3Ki design.

[1] University of Cologne, Faculty of Medicine and Cologne University Hospital, Department I of Internal Medicine; Center for Integrated Oncology Aachen, Bonn, Cologne, Düsseldorf, Cologne, Germany. [2] CECAD Cologne Cluster of Excellence on Cellular Stress Responses in Aging-Associated Diseases, Cologne, Germany. [3] Center for Molecular Medicine Cologne, Cologne, Germany. [4] Facultad de Química, Departamento de Física y Química Teórica, Universidad Nacional Autónoma de México, Mexico City, Mexico. [5] These authors jointly supervised this work: Rocio Rebollido-Rios, Günter Krause. ✉email: rocio.rebollido-rios@uk-koeln.de; guenter.krause@uk-koeln.de

The treatment of B lymphoid malignancies has been revolutionized by targeting kinases involved in B cell receptor (BCR) signaling, which include the phosphoinositide-3-kinase (PI3K) isoform δ[1]. Thus, disrupting the activity of p110δ for these indications resulted in the first clinical approvals of PI3K inhibitors (PI3Ki)[2]. Idelalisib, the first clinically applied PI3Ki, is an effective treatment of chronic lymphocytic leukemia (CLL)[3], but limited by immune-related adverse events and resistance[4,5]. Among targeted agents, PI3Ki therefore currently plays a less prominent role in the treatment of CLL compared to BTK inhibitors and BH3 mimetics[6].

PI3Ks catalyze the phosphorylation of the membrane constituent phosphoinositide-(4,5)-biphosphate (PIP2) to yield phosphoinositide-(3,4,5)-triphosphate (PIP3)[2]. By interaction of their pleckstrin-homology (PH) domains with PIP3, downstream mediators are recruited to the plasma membrane, where they are activated. Among these, the serine threonine kinase AKT serves as an important signaling hub and AKT phosphorylation is commonly used to monitor PI3K activity in cells. The four class I PI3K catalytic subunit isoforms have distinct though overlapping roles in pivotal physiological cell functions as well as in oncogenic signaling[7]. In this regard, the high frequency of activating point mutations in p110α, e.g., H1047R, in solid tumors points to an important targetable oncogenic potential[8,9]. In contrast, the similar E1021K substitution in p110δ occurs as a rare germline mutation that causes activated PI3K delta syndrome (APDS)[10]. Nevertheless, we employed this mutation to generate a p110δ-dependent mechanistic cellular model for PI3Ki assessment. The catalytic PI3K subunit p110δ further differs from p110α by preferential expression in hematopoietic tissues and physiological roles in antigen-receptor signaling and immune responses[11]. Since p110δ is overexpressed and constitutively activated by external stimuli, including the BCR, the survival of CLL cells, as well as their cytokine secretion and chemotaxis, can be targeted by idelalisib[12,13]. While genetic inactivation and isoform-selective pharmacologic targeting have largely explained the functional involvement of p110δ, these findings can be corroborated and refined by a chemical genetic approach, in which idelalisib-inhibited cell functions are rescued by a resistance mutation, such as the newly identified I777M substitution.

Among pharmacological PI3Ki, multi-targeted substances can be distinguished from p110δ-selective molecules that interact with a cleft between W760 and M752 of p110δ[14]. Consequently, mutation of M752 in p110δ leads to resistance against propeller-shaped p110δ-selective PI3Ki, whereas binding of flat multi-targeted PI3Ki remains unaffected. Comparisons of p110δ apo and PI3Ki-bound structures demonstrated an induced fit of propeller-shaped inhibitor molecules into the mentioned cleft, which is designated as a selectivity or specificity pocket[15]. Apart from this mechanism, the isoform selectivity of PI3Ki is determined by the inner core of the binding site, including non-conserved regions and conserved affinity pocket residues not directly contacting ATP[16,17]. To create drug-resistant p110δ, we introduced the I777M substitution that is analogous to the I800M mutation in p110α, which causes resistance to a few tested multi-targeted PI3Ki[18]. In some instances, this substitution close to the catalytic K779 was compared to the I825V mutation that affects the so-called gatekeeper residue, which controls inhibitor access to a pre-existing cavity within the ATP-binding pocket of protein and lipid kinases[19]. The I777M substitution allows the combined structural and cell-based examination of its impact on the binding of diverse PI3Ki as a contribution to ligand design.

To elucidate the molecular pharmacology of p110δ, we used isogenic cell lines expressing variants of this PI3K isoform. As a starting point, the activating C-terminal E1021K mutation enabled the development of a sensitive, isoform-selective mechanistic cellular model that we subsequently used for comparative high-throughput PI3Ki assessment. In addition, we employed the affinity pocket mutation I777M in p110δ, which caused resistance to idelalisib and thus provided the opportunity to explore the engagement of p110δ in BCR-dependent and oncogenic cell functions. The I777M mutation also served as a probe for p110δ interactions with diverse PI3Ki chemotypes in the aforementioned cell line-based assay system. The results of this assessment were related to structural investigation using molecular dynamics (MD) simulations.

## Results

**Oncogenic properties of p110δ.** As a prerequisite for PI3Ki characterization via cell viability, we examined the oncogenic potential of wildtype (wt) and mutant p110δ in the untransformed murine pro-B cell line BaF3, which requires interleukin 3 (IL-3) for growth under normal cell culture conditions, but can be rendered IL-3-independent by expression of different tyrosine kinases[20] or p110α-H1047R[21]. For this purpose, BaF3 cells were retrovirally transduced to express variants of p110α and p110δ at approximately equal levels in BaF3 cells (Supplementary Fig. 1a). Under IL-3 withdrawal, the viability of these constructs revealed the degree of factor-independence owing to overexpression and point mutations (Supplementary Fig. 1b). Expression of p110δ or p110α increased the percentages of IL-3-independent cells from 6 to ~20%. The E1021K or H1047R mutations augmented IL-3-independence to 30 or 50%, respectively, and further to above 50% in combination with binding pocket mutations in both isoforms. While in p110α, the main increase was due to the H1047R mutation, p110δ on its own showed slightly higher transformation capacity than p110α and only a slight increment with the E1021K mutation. To further explore the oncogenic potential of PI3K isoform variants, we used a different oncogenicity assay, namely anchorage-independent growth of murine fibroblasts[22]. In NIH3T3 cells the formation of colonies with an area of more than 1200 μm$^2$ in soft agar was increased approximately threefold by each, p110δ overexpression and the I777M mutation, which together reached only half of the effect of p110α-H1047R (Supplementary Fig. 1c).

To apply p110 isoform-dependent BaF3 cells for PI3Ki assessment, we modified the assay procedure to include a pre-incubation period of several days in the absence of IL-3. This combined selection for vector-encoded antibiotic resistance and factor-independent growth led to higher percentages of IL-3-independent viability and to a reduced difference between isogenic BaF3 cells with only E1021K or with additional I777M mutation (Fig. 1a). Subsequently, we investigated the concentration-dependent inhibition by idelalisib of the IL-3-independent growth of isogenic BaF3 cells expressing wt and mutants of p110δ (Fig. 1b). BaF3 cells expressing p110δ-wt showed higher sensitivity to idelalisib than parental BaF3 cells grown in IL-3-complemented medium. The activating E1021K mutation caused a further increase in sensitivity to idelalisib, resulting in a cellular half maximal inhibitory concentration (IC$_{50}$) in the order of 100 nM. To prove that we had generated a PI3K isoform-selective system for PI3Ki assessment, we used transduced BaF3 cells that express p110δ-E1021K or p110α-H1047R for determining the cellular sensitivity of three additional PI3Ki with known differences in efficacy and isoform selectivity (Fig. 1c). While duvelisib showed a strong preference for inhibiting p110δ, alpelisib affected preferentially the growth of p110α-dependent BaF3 cells and copanlisib led to strong and apparently equal inhibition of the cells transformed by both PI3K isoforms. These examples show that monitoring the cell viability of PI3K isoform-dependent BaF3 cells can serve as a tool for

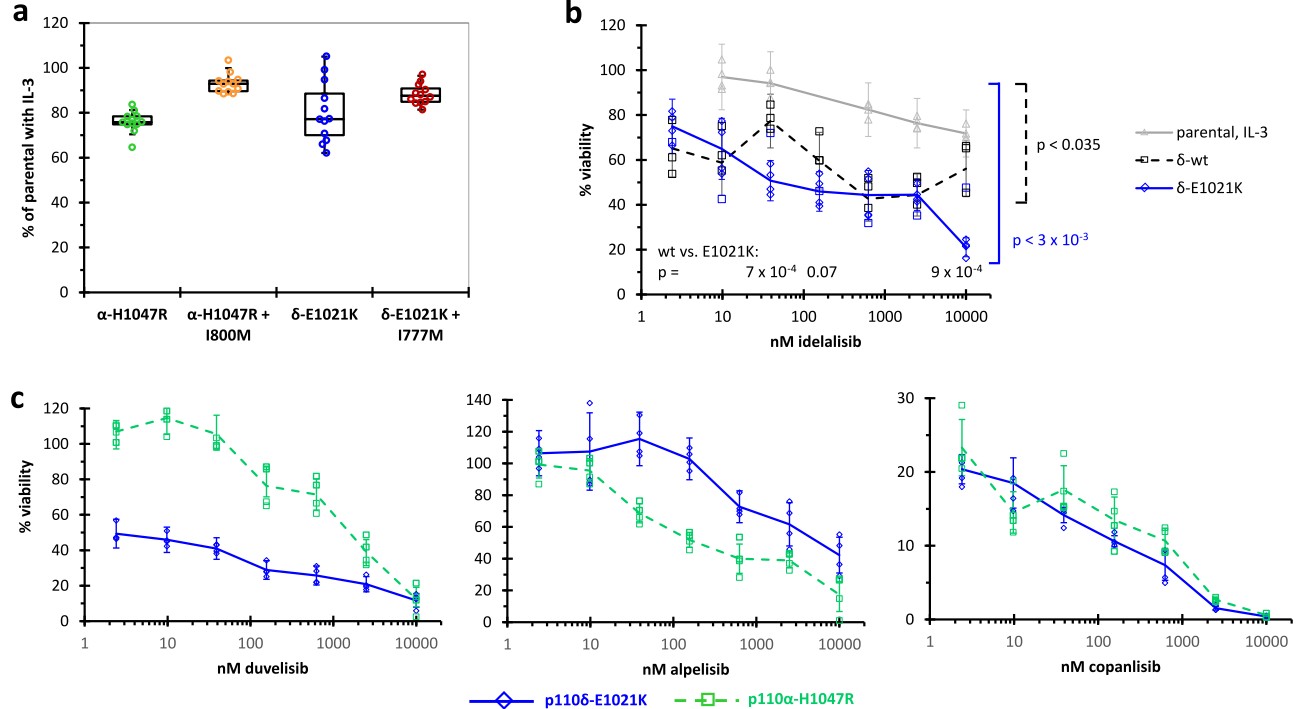

**Fig. 1 IL-3-independent growth and PI3Ki sensitivity of isogenic BaF3 cells.** BaF3 cells were retrovirally transduced to stably express the wild type or the indicated mutants of p110α and p110δ. After pre-incubation in media without IL-3 for several days, cell viability after four days in the absence of IL-3 was determined by bioluminescence measurements of ATP amounts with the CTG assay on 384-well plates. The viability of different isogenic BaF3 cells in the absence of IL-3 is expressed as a percentage of IL-3-complemented parental cells. P values for differences to parental BaF3 cells and indicated comparisons were determined by two-sided, unpaired T-tests. **a** IL-3-dependent growth of BaF3 cells expressing p110α and p110δ with C-terminal activating mutations alone or combined with additional binding pocket mutations was recorded. Per cell type, the box plots represent twelve measurements from two independent experiments. **b** The concentration-dependent cytotoxicity of idelalisib for isogenic BaF3 cells was determined under IL-3 deprivation for BaF3 cells expressing p110δ-wt or p110δ-E1021K and for parental BaF3 cells in the presence of IL-3. Means and single values of a representative quadruplicate measurement among four repetitions are shown. **c** The concentration-dependent responses to the PI3Ki duvelisib, alpelisib, and copanlisib were recorded in BaF3 cells expressing p110α-H1047R or p110δ-E1021K. Representative experiments were taken from three or four repetitions, respectively. Error bars indicate standard deviations among quadruplicate measurements.

comparatively assessing the potency and isoform selectivity of PI3Ki in a cell-based system that is amenable to high-throughput measurements.

**PI3Ki profiling by p110δ-dependent BaF3 cells.** To demonstrate their usefulness for PI3Ki assessment, we employed BaF3 cells expressing p110δ-E1021K for characterizing a larger set of structurally diverse PI3Ki. Owing to the common distinction of isoform-selective and multi-targeted PI3Ki[14], we classified these PI3Ki according to published biochemical isoform selectivity (Supplementary Fig. 2 and Supplementary Table 1). Multi-targeted PI3Ki with less than 14-fold selectivity to one than the other isoforms were subdivided into pan-class PI3Ki and dual PI3K/mTORi according to the presence or absence of potency against mTOR. The sensitivity of isogenic p110 isoform-dependent BaF3 cells to a collection of PI3Ki at concentrations from 2 nM to 10 μM was assessed by viability assays on 384-well plates.

Since isogenic BaF3 cells distinguished the isoform selectivity of example PI3Ki (Fig. 1c), we determined the ratios of cellular $IC_{50}$ values for p110δ and p110α of 27 PI3Ki (Fig. 2a). In a ranking according to isoform selectivity, p110δ- and p110α-selective PI3Ki flanked a range of multi-targeted PI3Ki, among which ZSTK474 exhibited high selectivity for p110δ. At the extremes, duvelisib and serabelisib showed 2660-fold or 3.8-fold cellular selectivity for p110δ or p110α, respectively. Overall, a median of 3.6-fold higher cellular than biochemical selectivity for

p110δ corresponded to enhanced sensitivity of p110δ-dependent BaF3 cells to this isoform. The present assessment of a PI3Ki library proved the capability of isogenic BaF3 cells to resolve the isoform selectivity of PI3Ki and to translate biochemical to cellular PI3Ki isoform selectivity.

Drug sensitivity testing using isogenic BaF3 cells showed a wide range of responses among the examined PI3Ki with pronounced differences between p110δ- and p110α-dependent BaF3 cells (Fig. 2b and Supplementary Table 2). Regarding p110δ as a relevant drug target in hematological malignancies, biochemical and cellular PI3Ki potencies of the PI3Ki in the present collection were largely in agreement (Supplementary Fig. 3a, b). A ranking of the examined PI3Ki by their cytotoxicity to BaF3 cells expressing p110δ-E1021K, indicated several substances with higher potencies than the clinically used and prototypic p110δ-selective PI3Ki idelalisib, namely nine multi-targeted and the p110δ-selective PI3Ki duvelisib and AMG319. In summary, BaF3 cells expressing p110δ-E1021K sensitively indicate differences in cellular PI3Ki potencies that generally relate well with reported biochemical potencies.

In addition to engineered BaF3 cells, we used the activated B cell-like (ABC) diffuse large B cell lymphoma (DLBCL) cell lines TMD8 and HBL-1 as tools for PI3Ki characterization (Fig. 2b). TMD8 cells responded considerably less to p110δ-selective PI3Ki than p110δ-dependent BaF3 cells, but their sensitivity to multi-targeted and p110α-selective PI3Ki followed a similar ranking (Supplementary Fig. 3c, d). HBL-1 cells showed at least 20-fold

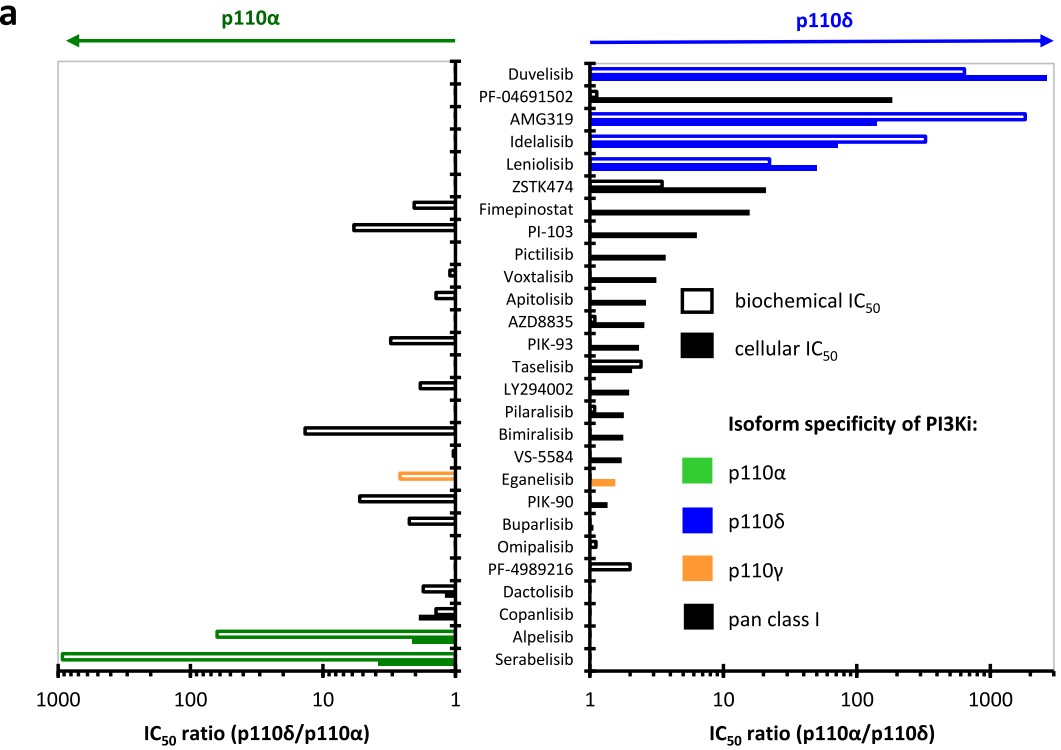

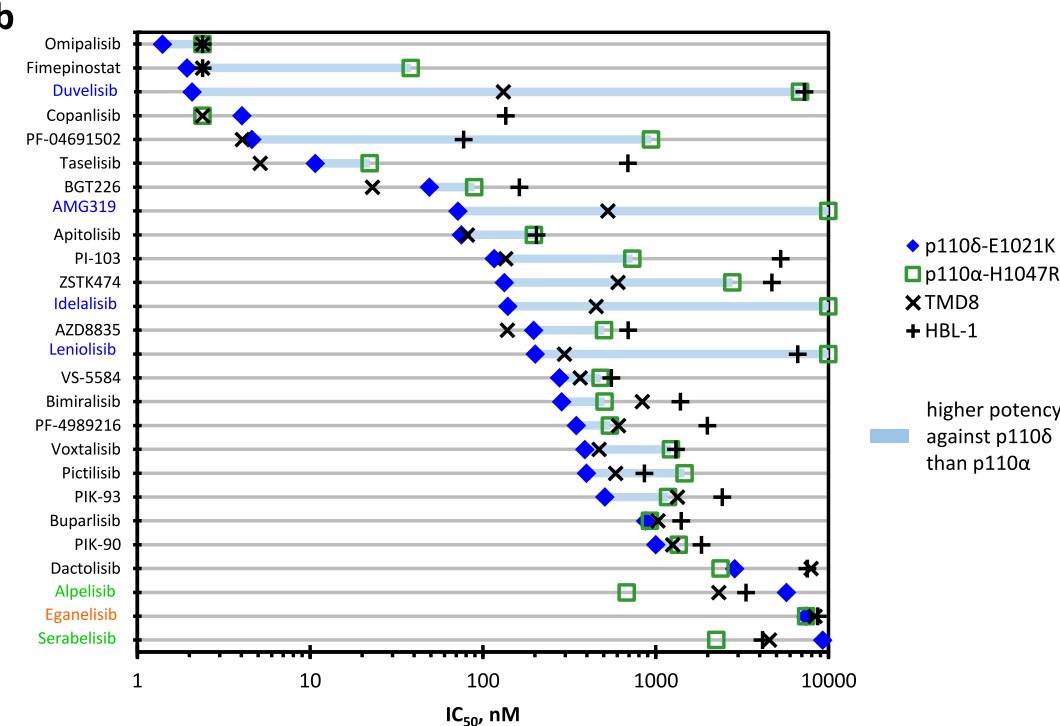

**Fig. 2 Cellular isoform selectivity and potency of structurally diverse PI3Ki.** The efficacy of a collection of PI3Ki was assessed in concentration-dependent viability assays with BaF3 cells expressing p110δ-E1021K or p110α-H1047K. Presented data are restricted to substances that show lower cellular $IC_{50}$ values than 10 μM against at least one of the investigated p110 isoforms. **a** For isoform selectivity profiling, PI3Ki were ranked according to the ratios of cellular $IC_{50}$ ratios of p110α/p110δ and compared with the corresponding ratios from published biochemical $IC_{50}$ values. Cellular and biochemical $IC_{50}$ ratios are presented by empty and filled bars, respectively. **b** The investigated PI3Ki were ranked according to their potency against BaF3 cells expressing p110δ-E1021K and compared with the efficacy of the same PI3Ki against p110α-H1047R and the DLBCL cell lines TMD8 and HBL-1. Compound names of pan-class I PI3Ki are printed in black. Blue, green, and orange print indicates PI3Ki selectivity for p110δ, p110α, and p110γ, respectively.

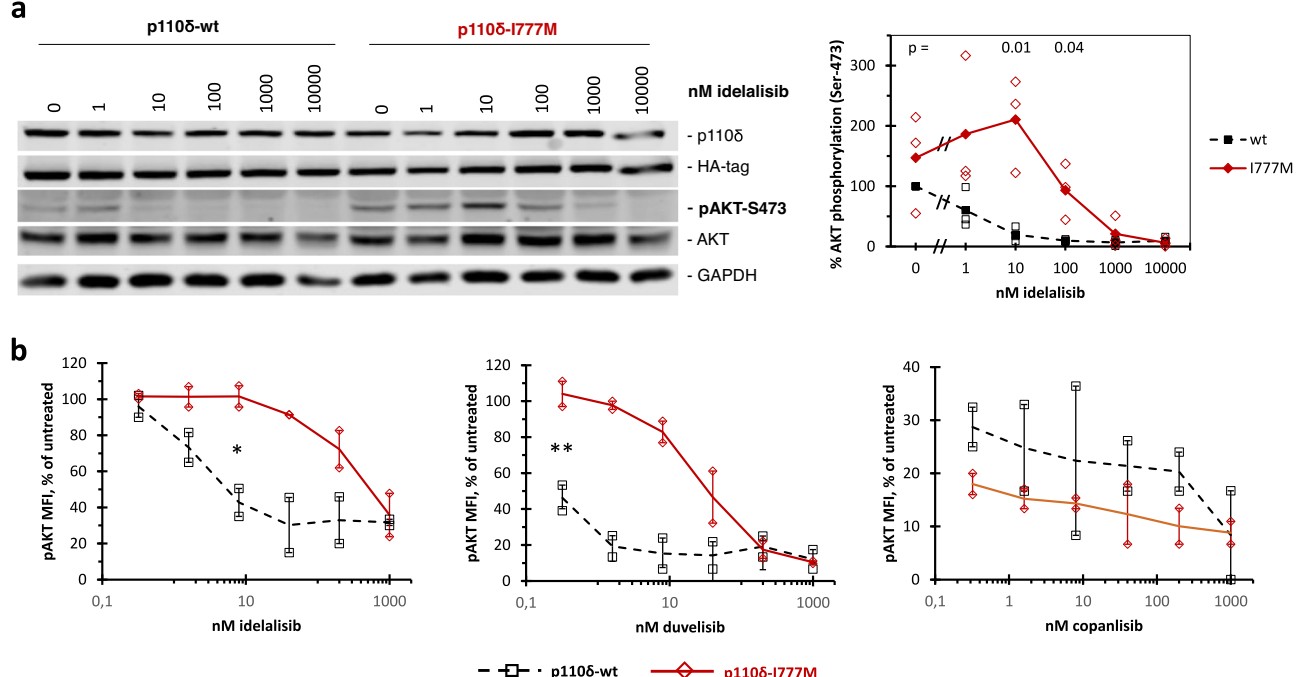

**Fig. 3 Resistance to idelalisib owing to the I777M mutation in p110δ.** Ramos cells were retrovirally transduced and selected to stably express wild type or mutant p110δ. **a** Transduced Ramos cells were stimulated with anti-IgM and treated with the indicated concentrations of PI3Ki prior to analysis of AKT phosphorylation at serine 473 in immune blots. The Western blots shown are representative of three independent experiments that also served for quantitation. Fluorescence signals were normalized to the corresponding total AKT and GAPDH references and expressed as percentages of the (anti-IgM stimulated) untreated wild-type control. **b** Concentration-dependent effects of different PI3Ki on the AKT phosphorylation at S473 in anti-IgM-stimulated Ramos cells were determined by phospho-specific flow cytometry. The average mean fluorescence intensity values of two independent measurements are shown. *$p < 0.05$; **$p < 0.01$.

lower sensitivity than TMD8 cells to all examined p110δ-selective PI3Ki and a few multi-targeted PI3Ki. Among control substances, TMD8 cells showed lower cellular potencies to inhibitors of BTK than p110δ-dependent BaF3 cells (Supplementary Fig. 4a, b). In summary, p110 isoform-dependent BaF3 cells combine sensitive isoform-selective PI3Ki assessment and high biological relevance.

**Idelalisib resistance caused by a binding pocket mutation.** We investigated the I777M mutation in p110δ, because the analogous mutation in p110α mediated resistance to some PI3Ki[18]. For this purpose, we transduced malignant B cell lines for stable expression of mutant and wt p110δ at comparable levels and monitored AKT phosphorylation as an indirect measure of PI3K activity. In Ramos cells, which can be activated by clustering the BCR with anti-IgM, AKT phosphorylation at S473 more than doubled after treatment with anti-IgM (Supplementary Fig. 5a). Transgene expression and pAKT levels were also followed in BaF3 cells with and without IL-3 stimulation (Supplementary Fig. 5b). Compared to wt, AKT phosphorylation was increased by the I777M mutation in both cell lines, while the gatekeeper mutation I825V had the opposite effect.

Concentration-dependent immune blot analysis of anti-IgM-stimulated Ramos cells indicated that the I777M mutation maintained AKT phosphorylation in the presence of idelalisib (Fig. 3a). Compared to wt, p110δ-I777M significantly rescued AKT phosphorylation at 10 and 100 nM idelalisib. A similar effect of the I777M mutation on pAKT levels was observed in BaF3 cells (Supplementary Fig. 6b). In contrast, the gatekeeper mutation I825V, which strongly reduced AKT phosphorylation in activated Ramos and BaF3 cells, enhanced the sensitization to idelalisib owing to p110δ overexpression (Supplementary Fig. 6a, c). In NIH3T3 cells, which are essentially devoid of endogenous

p110δ expression, forced expression of p110δ-wt led to increased sensitivity to idelalisib (Supplementary Fig. 7a). This sensitization to idelalisib owing to overexpression of p110δ was less pronounced in Ramos and BaF3 cells with increasing endogenous p110δ expression (Supplementary Fig. 7b, c). To extend concentration-dependent signaling analyses to additional PI3Ki, anti-IgM-stimulated Ramos cells were also examined by phospho-specific flow cytometry (Fig. 3b). Idelalisib, duvelisib, and copanlisib differentially inhibited the AKT phosphorylation in Ramos cells expressing p110δ-wt. Compared to wt, Ramos cells expressing p110δ-I777M were approximately 100 times less sensitive to idelalisib and duvelisib and equally sensitive to copanlisib. Thus, according to AKT phosphorylation, the I777M mutation in p110δ mediated strong resistance to idelalisib.

**Rescue of cell functions in the presence of idelalisib.** Maintained AKT phosphorylation in the presence of idelalisib owing to the I777M mutation in p110δ enabled chemical genetic evaluation of the functions of malignant B cells. For this purpose, we examined the concentration-dependent inhibition by idelalisib of cytokine secretion in Ramos cells with activated BCR signaling (Fig. 4a). Overexpression of p110δ-wt augmented the anti-IgM-induced secretion of CCL3, but not of CCL4 and TNFα compared to the vector control (Supplementary Fig. 8a). The I777M mutation further increased the secretion of CCL3 but not of CCL4 and TNFα compared to wt in a similar manner, and, importantly, maintained the secretion of all three cytokines in the presence of idelalisib (Fig. 4a). In contrast to metabolic activity (Supplementary Fig. 8b), the I777M mutation reconstituted cytokine secretion of Ramos cells in the presence of idelalisib.

In addition, we investigated the transwell migration to CXCL12 of BaF3 cells expressing wt or mutant p110δ. Chemotaxis to

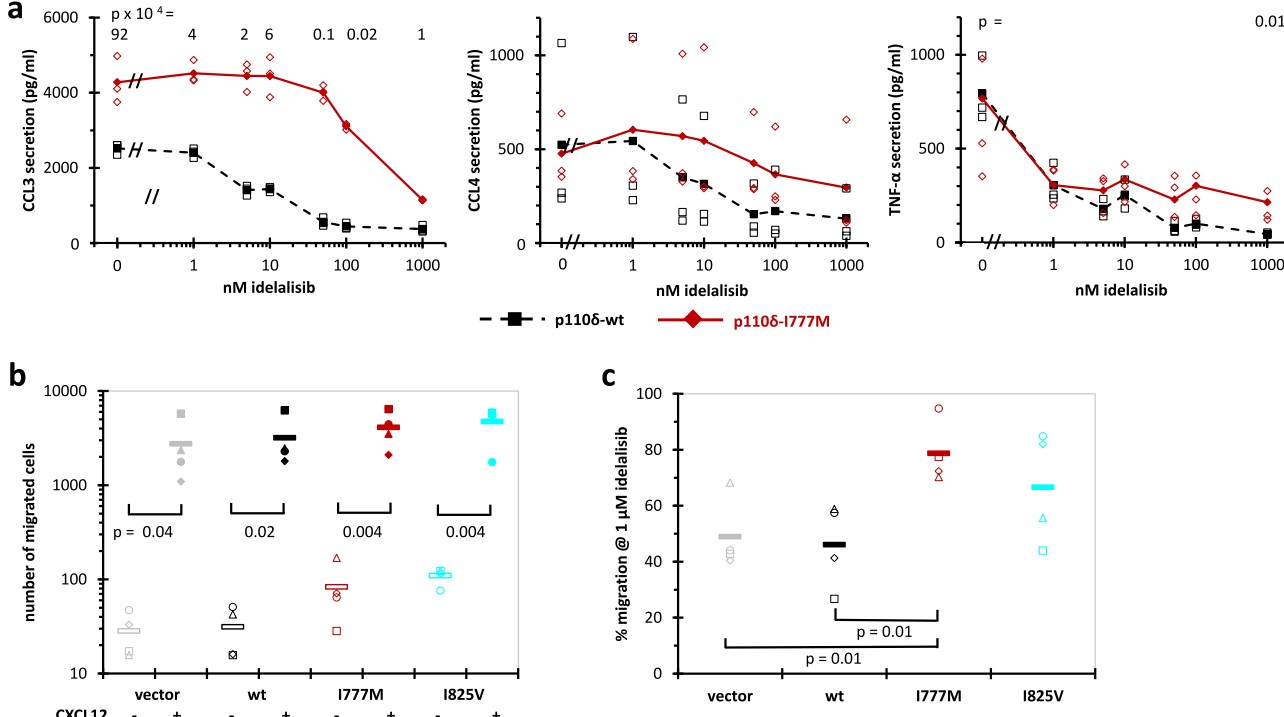

**Fig. 4 Rescue of cell functions in the presence of idelalisib.** Ramos (**a–c**) or BaF3 (**d**) cells were retrovirally transduced and selected to stably express wild type or mutant p110δ. **a** Ramos cells expressing p110δ or p110δ-I777M were used to determine the anti-IgM induced secretion of the cytokines CCL3, CCL4, and TNFα in the presence of PI3Ki. Cytokine levels in culture supernatants were determined by ELISA. For each cytokine, three independent assays were performed. **b**, **c** The chemotaxis of isogenic BaF3 cells was examined in transwell migration assays. The numbers of migrated BaF3 cells expressing p110δ variants were compared in the absence and presence of the chemo-attractant CXCL12, as indicated by empty and filled symbols (**b**). Percentages of cells that migrated to CXCL12 in the absence or presence of 1 μM idelalisib were calculated from parallel experiments with passage-matched samples (**c**). Means and single values of four biological replicates are shown. *P* values were determined by two-sided unpaired *T*-tests.

CXCL12 increased the number of migrating cells approximately one-thousand-fold compared to spontaneous migration (Fig. 4b). BaF3 cells expressing p110δ-I777M showed approximately threefold higher spontaneous migration than wt or vector controls. Compared to wt, p110δ-I777M significantly maintained the chemotaxis of BaF3 cells to CXCL12 in the presence of 1 μM idelalisib (Fig. 4c). Rescue of cytokine secretion and cell migration in the presence of idelalisib by the I777M mutation documented p110δ engagement in B cell functions involved in the micro-environmental dialog.

**Mutation-mediated resistance to diverse PI3Ki.** For further characterization of the I777M substitution, we took advantage of our newly developed p110δ-selective cell line model harboring the activating E1021K mutation. Indeed, these p110δ-dependent BaF3 cells indicated viability as a further crucial cell function that can be reconstituted by the I777M mutation for the evaluation of PI3Ki resistance (Fig. 5). The I777M mutation significantly rescued the viability of isogenic BaF3 cells at several concentrations of idelalisib, duvelisib, umbralisib, and ZSTK474, but not copanlisib (Fig. 5a, b). In a larger set of PI3Ki arranged according to decreasing cellular PI3Ki potency, we recorded changes of the cellular IC$_{50}$ in p110δ-dependent BaF3 owing to the I777M mutation (Fig. 5c). While this substitution caused prominent resistance mainly to p110δ-selective PI3Ki, the analogous I800M mutation in p110α showed strong resistance to most multi-targeted PI3Ki, particularly the three most potent ones (Supplementary Fig. 9a). In addition, we ranked PI3Ki according to the strength of mutation-mediated resistance expressed as IC$_{50}$ ratios (Supplementary Fig. 9b, c and Supplementary Table 3). The maximal observed resistance owing to p110δ-I777M or p110α-

I800M was 25- or 480-fold with duvelisib or copanlisib, respectively. Remarkably, p110δ-I777M led to sensitization to some PI3Ki, predominantly those with additional activity against mTOR. The I777M mutation in p110δ-dependent BaF3 cells did not result in apparent resistance to other drugs than PI3Ki (Supplementary Fig. 4c, d). Individual PI3Ki often showed opposite positions in the resistance rankings determined for p110δ and p110α. Apart from p110δ-selective PI3Ki, the I777M mutation in p110δ caused strong resistance to ZSTK474, but none to copanlisib. Together with the availability of structural data (Supplementary Table 4), these observations defined the selection of substances for the analysis of binding pocket interactions by molecular dynamics simulations, namely of idelalisib as representative of p110δ-selective PI3Ki in comparison with the pan-class I PI3Ki ZSTK474 and copanlisib.

**Structural and dynamical changes in p110δ caused by the I777M substitution.** To explain our pharmacological observations at the molecular level, we investigated the structural basis of the interaction of PI3Ki with p110δ. For this purpose, we performed MD simulations of structural models of human p110δ-wt and −I777M with and without small molecule inhibitors bound (Supplementary Table 5). According to backbone fluctuations, the I777M substitution increased the overall flexibility of the p110δ apo-structure, particularly in the P-, catalytic, and activation-loops and the hinge region (Supplementary Fig. 10a).

Comparisons of apo and idelalisib-bound p110δ-wt (Supplementary Movies 1, 2) showed an induced fit of the quinazoline moiety of idelalisib into the specificity pocket between M752 and W760, which did not form in p110δ-I777M (Fig. 6a). Idelalisib binding to p110δ-wt was reinforced by a very stable hydrogen

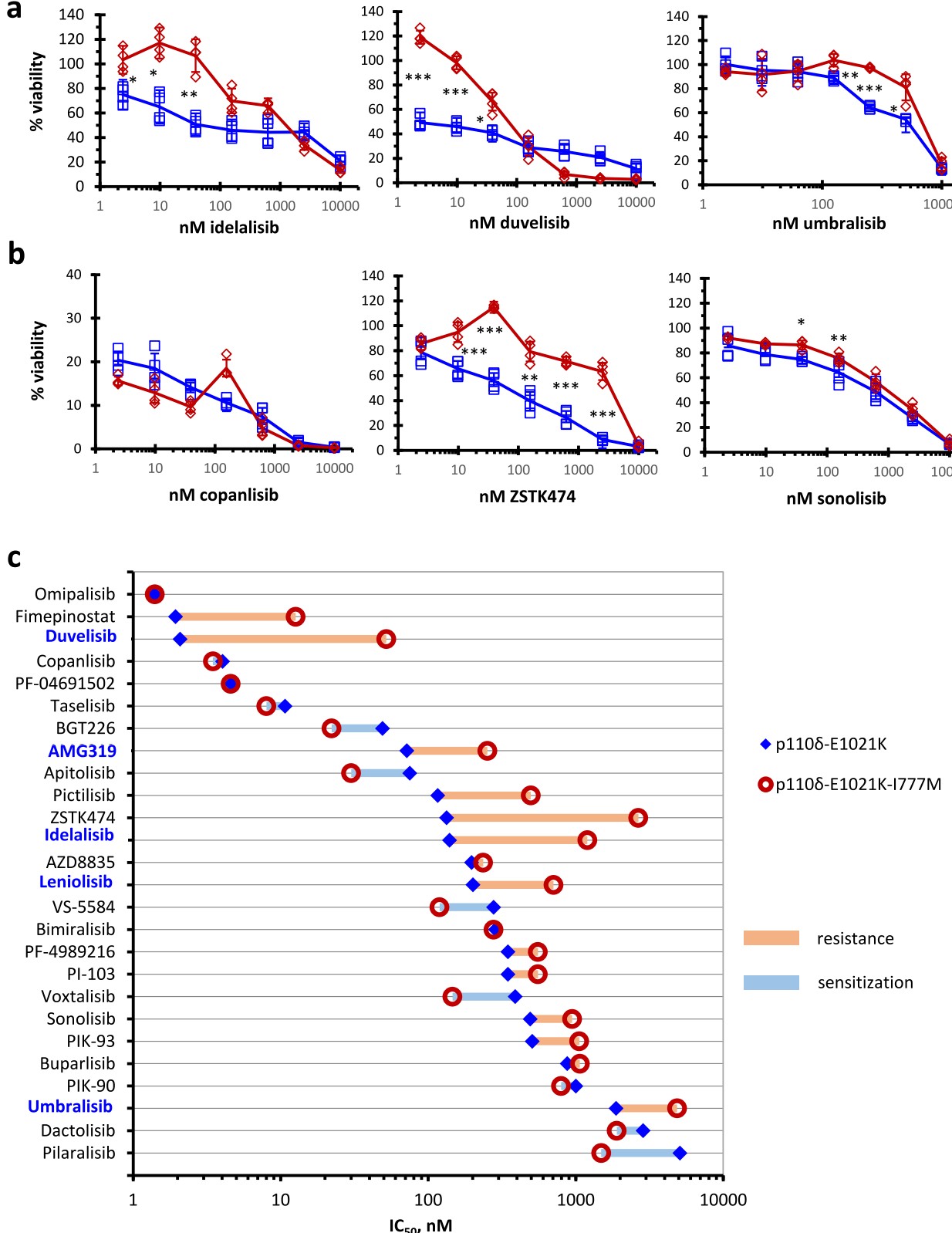

bond between V828-NH in the hinge region and a nitrogen from the purine moiety of idelalisib, which was exhibited all over the trajectory (Supplementary Fig. 10b, d). In addition, there were water-mediated hydrogen bonds between idelalisib and residues M900 and I910 of the hydrophobic region II adjacent to the activation loop of p110δ-wt (Supplementary Fig. 10b, d). In p110δ-I777M, however, a conformational rearrangement of the

ATP-binding pocket occurred, that resulted in a steric clash of the M777 side chain with idelalisib as positioned in p110δ-wt (Fig. 6b, c). Therefore, idelalisib adopted an altered position and came close to different residues in p110δ-I777M than, in p110δ-wt, namely Y813 as well as S831 and T833 of the hinge region (Supplementary Fig. 10c). The altered conformation of p110δ-I777M resulted in the joint shift of M752 and W760 side chains

**Fig. 5 Impact of the resistance mutation I777M in p110δ on the cellular efficacy of diverse PI3Ki.** BaF3 cells were retrovirally transduced to stably express the p110δ-E1021 without or with additional I777M mutation for CTG assays after IL-3 withdrawal. **a, b** Concentration-dependent responses to the p110δ-selective PI3Ki idelalisib, duvelisib, and umbralisib (**a**) and to the pan-class I PI3Ki copanlisib, ZSTK474, and sonolisib (**b**) were recorded in isogenic BaF3 cells. Representative experiments of four repetitions are shown. Error bars indicate standard deviations among quadruplicate measurements and asterisks significant differences between wt and I777M. *$p < 0.05$; **$p < 0.01$; ***$p < 0.001$. **c** PI3Ki that showed cellular $IC_{50}$ below 5 µM with BaF3 cells expressing p110δ-E1021K were arranged according to decreasing potency and compared with the corresponding $IC_{50}$ values of isogenic cells with additional I777M mutation. P110δ-selective PI3Ki are printed in blue and dual PI3K-mTORi in italics. The red and blue bars indicate resistance or sensitization, respectively.

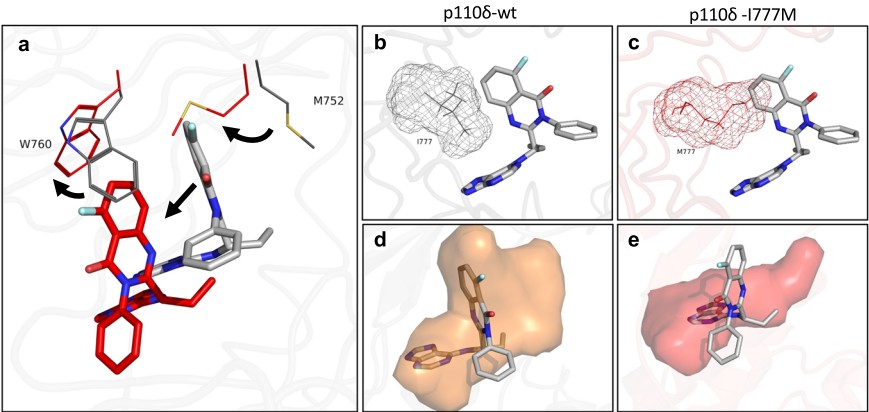

**Fig. 6 Idelalisib binding to p110δ-wt and −I777M.** Representative structures of p110δ-wt (gray) and p110δ-I777M (red) were derived from MD simulations. **a** The side chains of residues M752 and W760 are shown as lines. Sticks representing idelalisib are colored gray or red in interaction with p110δ-wt or −I777M, respectively. Arrows indicate shifts in the positions of specificity pocket residues and displacement of the idelalisib molecule owing to the I777M substitution. **b, c** Mesh representation of residue 777 highlighting the space occupied by isoleucine (**b**) or methionine (**c**). **d, e** Binding pocket volume of p110δ-wt with 643 Å³ (**d**) and p110δ-I777M with 357 Å³ (**e**) colored in orange and red, respectively.

and thus prevented the opening of the specificity pocket, which resulted in the displacement of idelalisib (Fig. 6a and Supplementary Movie 2). Accordingly, the volume of the idelalisib binding pocket was reduced from 643 Å³ in p110δ-wt to 357 Å³ in p110δ-I777M (Fig. 6d, e). Disruption of the specificity pocket in p110δ-I777M was reflected by significant changes compared to p110δ-wt in additional structural features, e.g., distances, angles, and binding pocket surface area (Supplementary Fig. 10d–f). Overall, MD simulations provided clear-cut evidence that the I777M substitution led to a conformational change that caused idelalisib resistance by preventing the formation of the specificity pocket between M752 and W760.

Despite its classification as pan-class I PI3Ki according to isoform selectivity, ZSTK474 was subject to I777M-mediated resistance in cell-based assays. MD simulations of ZSTK474 binding to p110δ-wt suggested that the benzimidazole group was accommodated in a pocket formed by residues I777 and K779 that was smaller in the absence of the inhibitor (Fig. 7a and Supplementary Movie 3). One of the morpholino groups was positioned close to the hinge region (residues I825–V828) with the kinase N- and C-lobes approaching each other. One fluor atom of the difluoromethyl group pointed towards K779 and the other lined up with one of the two morpholino groups, rotated by 90°, both projecting out of the pocket and highlighting a non-planar ZSTK474 geometry (Fig. 7a and Supplementary Movie 4). A hydrogen bond was established between the nitrogen of the benzimidazole of ZSTK474 and the amino group of K779 (Supplementary Fig. 11c, d). The I777M substitution, however, prevented ZSTK474 from entering into the pocket, since the opening between I777 and K779 was no longer available (Fig. 7b). Though smaller than the specificity pocket induced upon idelalisib binding, the ZSTK474 binding pocket volume was larger in wt than mutant p110δ. Compared to wt, the difluoromethyl group changed the orientation facing

upwards to M752 and P758 (Supplementary Fig. 11a). ZSTK474 moved closer to K779 and D787 with one morpholino group approaching T833. Several features in the molecular geometry of ZSTK474 binding indicated significant differences between p110δ-wt and −I777M (Supplementary Fig. 11b). For ZSTK474, the I777M substitution led to resistance due to conformational changes that mainly involved residues of the affinity pocket.

To find out why the I777M mutation did not mediate resistance to copanlisib in cellular assays in contrast to idelalisib and ZSTK474, we performed MD simulations of this PI3Ki molecule bound to p110δ-I777M. The position of copanlisib extended from residues capable of forming a specificity pocket via the adenine and affinity pocket towards N898 and D911 at the mouth of the ATP-binding pocket and close to the activation loop (Fig. 7c and Supplementary Movie 5). Compared to idelalisib and ZSTK474, copanlisib exhibited a less rotatable and flexible geometry. Hence, even though M777 was within 3 Å to all three PI3Ki, it obstructed the binding of idelalisib and ZSTK474, but not copanlisib.

In addition to M777, residues M752, V827, T833, M900, and I910 were common contact residues of the three inhibitors. ZSTK474 exclusively contacted residues K779, D787, E826, and T833 of the extended affinity pocket and idelalisib residue S831. Taken together, our in silico findings show, how the I777M substitution precluded the binding of idelalisib and ZSTK474 to p110δ inducing a conformational rearrangement of the binding site, while copanlisib binding was not affected by the I777M mutation.

## Discussion
Using C-terminal activating mutations of p110, we developed a cell-based high-throughput system for the assessment of PI3Ki

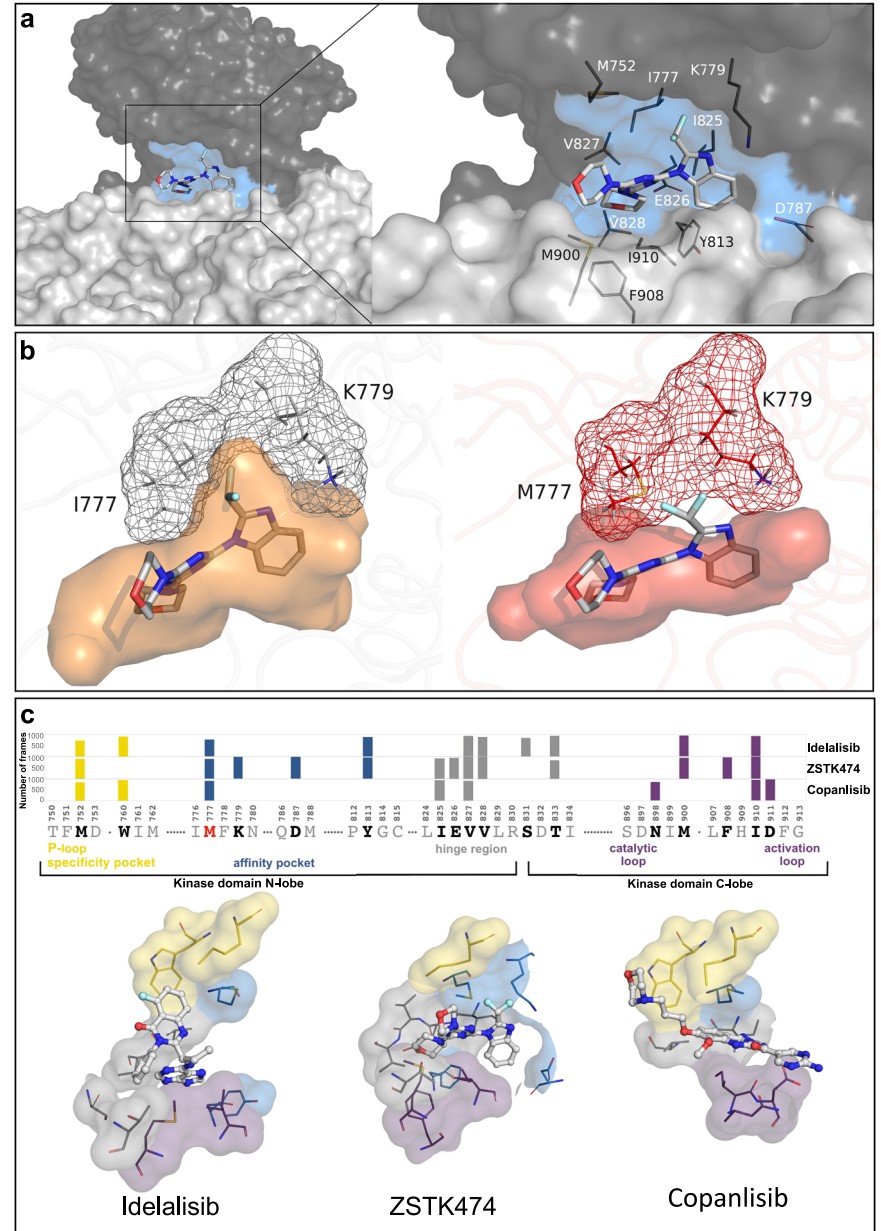

**Fig. 7 Impact of the I777M substitution on PI3Ki binding.** Representative structures were derived from MD simulations. **a** ZSTK474 accommodation in the binding cleft between the N-lobe (dark gray) and C-lobe (light gray) of the p110δ-wt kinase domain. The contact area is indicated by blue surface color and prominent residues are emphasized by stick presentation in the zoom-in. **b** Conformational changes induced by the I777M mutation affect ZSTK474 binding. The difluoromethyl group of ZSTK474 fits between residues I777 and K779 (gray mesh) of p110δ-wt, but not between M777 and K779 of mutant p110δ (red mesh). The color of the mesh around M777 indicates sterically hindered (red) or undisturbed (gray) inhibitor binding. Moreover, the I777M substitution reduces the size of the binding pocket (red), from which the ZSTK474 molecule is expelled. **c** Different molecular interaction of idelalisib, ZSTK474 and copanlisib with p110δ-I777M. In the bar plots on the top, binding pocket residues are arranged from N to C terminus and assigned to regions of the specificity pocket (yellow), adenine and affinity pocket (blue), excluding an extended hinge region (gray), as well as catalytic and activation loop (purple). The height of each bar represents the total number of frames where the corresponding residue was found within a 3 Å distance to the inhibitors out of 1000 from each simulation. Only residues found in more than 75% of the frames are shown. The residues displayed in the bottom panel correspond to those shown in the top panel and are located within the binding pocket of individual inhibitors. Each residue is assigned a region-specific color according to the bar plots. The inhibitors are represented as balls and sticks and the residues as lines. Of note, the enzyme ligand structures show disturbed interactions with idelalisib and ZSTK474 in contrast to copanlisib binding, that is not subject to mutation-mediated resistance. For these three PI3Ki, the residues within a 3 Å distance illustrate interaction with different binding pocket regions.

isoform selectivity and potency. Moreover, we exploited the affinity pocket mutation I777M to evaluate p110δ engagement in BCR-dependent and oncogenic cell functions and to provide an experimental basis for linking cell-based assessment with structural models of PI3Ki interactions (Fig. 8).

To generate a system for cell-based, p110δ-selective PI3Ki assessment, we took advantage of the transformation of BaF3 cells to factor-independence owing to the E1021K mutation in p110δ[10], similarly as previously performed with p110α-H1047R in a different cell line[18]. The IL-3 independence gained by

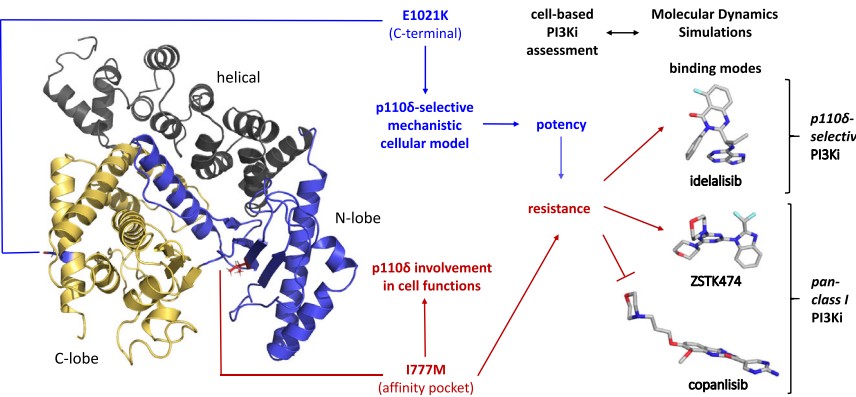

**Fig. 8 The PI3K isoform δ as a drug target.** An activating mutation (blue) in p110δ enables comparisons of the cellular potencies of diverse PI3Ki in viability assays. The affinity pocket mutation I777M (red) consistently mediates resistance to p110δ-selective PI3Ki, but only occasionally to pan-class I PI3Ki. In accordance with cell-based assays, MD simulations indicate resistance owing to the I777M substitution for idelalisib and ZSTK474, which project from the plain occupied by ATP, but not for the flat copanlisib molecule. MD simulations also show how the I777M substitution leads to conformational changes that can affect PI3Ki interactions with p110δ. Three-dimensional poses of PI3Ki interacting with p110δ-I777M were taken from the equally oriented structures in Fig. 7c.

transduction of BaF3 cells with p110δ and p110α controls was in concordance with the reported transformation potential of these molecules for fibroblasts[23,24]. The E1021K mutation in p110δ, similar to p110α-H1047R[21], oncogene fusions of tyrosine kinases[25] or myristoylation of AKT[26] secured high inhibitor sensitivity of the IL-3-independent viability of BaF3 cells. The ratios of $IC_{50}$ values obtained for the examined PI3Ki with isogenic p110δ- and p110α-dependent BaF3 cells spanned four orders of magnitude, which surpassed the resolution of isoform selectivity observed in an analysis of isoform-selective AKT inhibitors in a similar BaF3 cell-based system[26]. In contrast, the few cellular mechanistic models hitherto applied for PI3Ki assessment relied on the detection of AKT phosphorylation rather than cell viability[27,28]. For three PI3Ki in our study, the viability of p110δ-dependent BaF3 cells (Fig. 4a) showed similar concentration-response relationships as pAKT levels in Ramos cells with activated BCR (Fig. 3b) at approximately 10-fold lower sensitivity. Compared to a set of diverse cell lines with predominantly activated PI3K class I isoforms[27], p110δ- and p110α-dependent BaF3 cells provided a coherent assay system that potentially can be extended by BaF3 cells expressing mutationally activated p110β and p110γ. Another mechanistic cellular system for PI3Ki assessment, namely fibroblasts expressing myristoylated class I PI3K isoforms[28], uses more rigid PI3K activation than C-terminal activating mutations and a cell context that is less related to B cell malignancies than our newly established system for PI3Ki profiling via straightforward viability measurement. With these advantages compared to existing systems, our newly developed mechanistic model not only allowed the determination of isoform-selective cellular PI3Ki potencies, but also served to evaluate the effects of the affinity pocket mutation I777M on p110δ interactions with PI3Ki.

We introduced this mutation close to the catalytic K779 in p110δ to explore whether it renders the enzyme opaque to pharmacological inhibitors. Its characterization in Ramos cells revealed that, indeed the I777M substitution mediated resistance to idelalisib and, in addition, unexpectedly led to increased pAKT levels corresponding to a gain of function (Fig. 3a, b and Supplementary Fig. 5), whereas the analogous mutation in p110α moderately reduced PI3K activity[18]. Co-occurrence in p110δ of I777M and E1021K point mutations resulted in enhanced transforming potential, which is reminiscent of the combined effect of two kinase domain mutations in BTK[29]. The I777M substitution reconstituted pAKT levels at ~100-fold increased

idelalisib concentrations, which is similar to the dasatinib resistance mediated by the gatekeeper mutations in the BCR-associated kinases BTK or LYN and higher than the ibrutinib resistance owing to the C481S mutation in BTK that prevents irreversible inhibitor binding[30]. Although the I777M mutation at the affinity pocket of p110δ functionally impedes the efficacy of idelalisib and thus essentially could presage acquired resistance to this drug, whole exome sequencing (WES) did not detect recurrent mutations in p110δ in samples from patients with disease progression on idelalisib[5]. This is corroborated by the fact that specifically in the case of PI3Ki, it is notoriously difficult to generate drug-resistant cell lines through in vitro exposure over long periods of time[9], as well as by WES analysis of idelalisib-resistant TMD8 cells generated in this manner[31] and in an in vivo model using serial tumor transfer and treatment[32]. In contrast to clinically acquired mutations in the tyrosine kinases Abl[33] and BTK[34], resistance to idelalisib is mostly based on compensatory mechanisms, such as PTEN loss and up-regulation of PI3Kγ expression[31], upregulated IL-6 and PDGFR signaling[35], activation of IGF1R[32] or mutations that activate the MAPK signaling cascade[36]. In summary, the I777M functionally mediates resistance to idelalisib, leaving the intriguing question, of why this binding pocket mutation is not selected for in vivo, in contrast to similar mutations in tyrosine kinases.

The cell-based assay systems used in this study relied on the expression of p110δ variants in addition to endogenous wt p110δ. Since endogenous p110δ levels increased from NIH3T3 to Ramos and BaF3 cells, overexpression of p110δ increased in this order and led to correspondingly augmented sensitivity to idelalisib (Supplementary Fig. 7a–c). This sensitization by p110δ overexpression can be attributed to p110δ-selective PTEN activation by still unknown mechanisms, conceivably protein kinase activity of p110δ or compartmentalized signaling[37,38]. This effect can lead to dampened resistance to dual inhibitors of PI3K and mTOR and even amount to overall sensitization due to the concomitant blockade of an independent target further downstream in the same pathway (Supplementary Fig. 7d). Moreover, the contribution of p110δ overexpression to oncogenic transformation by far exceeded that of p110α, presumably owing to the pro-B cell nature of the parental BaF3 cells. Both, sensitization to inhibition of p110δ and biased p110δ dependence and, may have contributed to shifting the observed cellular isoform selectivity of PI3Ki towards p110δ compared to the biochemically determined values. Background expression of p110δ-wt did not disturb the

analysis of resistance, since PI3Ki treatment exerted selection pressure against it. Taking the mentioned limitations into account, the generated cell-based systems accurately indicated isoform-selective PI3Ki effects and resistance in a cell context.

The present PI3Ki assessment focused on the inhibition of p110δ as the predominant isoform in B cell malignancies and used a PI3Ki collection that included the five PI3Ki currently approved for clinical use and the irreversible PI3Ki sonolisib. For this library of structurally diverse PI3Ki, our newly established p110δ-dependent cell-based system showed lower potencies among p110δ-selective than multi-targeted PI3Ki in accordance with earlier observations[14]. Apart from p110δ-selective PI3Ki, the ranking of PI3Ki efficacy against p110δ-dependent BaF3 cells resembled that against DLBCL cell lines. TMD8 and HBL-1 cells with low and high p110α expression, respectively, corroborated the advantage of additionally targeting p110α, e.g., by copanlisib and AZD8835[39,40], which also was observed in malignant B cells from mantle cell lymphoma and CLL[41,42]. Overall, we created a resource of PI3Ki characterization that connected isoform-selective and disease-relevant cellular PI3Ki potencies for comparisons with patient-specific or other pre-clinical PI3Ki assessment, e.g., in CLL or DBLCL cells[43,44], and for the extension to the analysis of PI3Ki resistance.

Although apparently not involved in clinical idelalisib resistance, the I777M substitution provides an exquisite tool for dissecting the roles of PI3K isoforms in a cell context. The dependence of the cytokine secretion by Ramos cells on stimulation of the BCR and increased CCL3 production owing to p110δ overexpression attest to a prominent role of this PI3K isoform as a BCR-associated kinase. In particular, the concentration-dependent response to idelalisib of Ramos cells expressing wt and resistance mutant of p110δ was more directly translated from pAKT levels to CCL3 secretion than the corresponding responses to ibrutinib and dasatinib to activation of BTK and Src family kinases that had been investigated in analogous systems expressing wt and mutant BTK and LYN[30], indicating closer linkage of BCR-activated CCL3 secretion with the PI3K/AKT than the MEK/ERK or PLCγ/PKCβ signaling axis. Enhanced oncogenicity and rescue of directed transwell migration in the presence of idelalisib by the I777M mutation also proved engagement of p110δ in basal and CXCL12-dependent mobility of BaF3 cells, respectively. According to chemical genetic evaluation, p110δ was engaged in cell functions with importance for the microenvironmental dialog, e.g., cytokine secretion and chemotaxis, but also in p110δ-driven cell viability. The newly developed toolset for the functional evaluation of PI3K isoforms can be applied to models representing tumor cells as well as cell types of the tumor micro-environment, where BCR-associated kinases also play relevant roles[45].

Our cell-based PI3Ki assessment indicated that resistance due to the I777M mutation in p110δ mainly concerned p110δ-selective PI3Ki. For their prototypic representative, idelalisib, MD simulations allowed to directly follow induction of the specificity pocket upon ligand binding, as previously concluded from X-ray structures of co-crystals[15,46], and showed that the I777M mutation disturbed its formation via conformational changes that affected the position of M752. The most prominent structural change owing to the I777M substitution observed in the interaction of p110δ and idelalisib occurred distantly from the mutation site at this residue that is known to be involved in the binding of propeller-shaped PI3Ki[14], but also of leniolisib[28]. Similarly, the preceding non-conserved residue, F751 in p110δ, regulated isoform selectivity, as exemplified by the reciprocal mutations I771Y and Y778I in p110α and p110β, respectively[47]. Since the PI3K class I isoforms p110δ and p110β at this position bear aromatic residues that permit ligand binding, as opposed to aliphatic residues in p110α and p110γ, one might

speculate that the analog of the I777M mutation in p110β (I883M) would also cause resistance to propeller-shaped p110β-selective PI3Ki. In ZSTK474-bound p110δ, MD simulations showed a cryptic pocket between I777 and K779, the formation of which was prevented by the I777M substitution. In contrast, copanlisib bound equally well to p110δ-wt and p110δ-I777M, whereas the analogous I800M mutation in p110α mediated strong resistance. This striking difference between two isoforms in the effect of analogous mutations at a conserved affinity pocket residue is reminiscent of the role of the non-conserved residue mentioned above in isoform-specific drug affinity[47]. For three selected PI3Ki, cell-based assessment and MD simulations arrived at the same conclusions as to mutation-mediated PI3Ki resistance, implying the importance of projections from the main plain of PI3Ki molecules for their interactions with N-lobe regions of p110δ. Vice versa, the I777M mutation provides a straightforward model for virtual ligand design that can be easily tested experimentally using the described cell-based systems.

Taken together, mutations in p110δ enabled convenient cell-based assessment of isoform-selective PI3Ki efficacy as well as chemical genetic evaluation of p110δ-dependent cell functions and insights into the molecular interactions of PI3Ki with the binding cleft of p110δ.

## Methods

**Cell lines and cell culture.** Ramos, BaF3, and NIH3T3 cells were obtained from the German Collection of Microorganisms and Cell Cultures (DSMZ, Braunschweig, Germany). TMD8 and HBL-1 cells were a gift from Dr. Georg Lenz (University of Münster). Cells were maintained in DMEM or RPMI-1640 medium supplemented with 10% fetal bovine serum and penicillin-streptomycin. The parental BaF3 cells were supplemented with 10 ng/ml IL-3. All cells were cultured at 37 °C at 5 % $CO_2$.

**Expression constructs.** PCR products containing full reading frames of human p110δ or p110α were generated from plasmid templates (Addgene #34893, 12522, 12524) for cloning into the XhoI/SacII or XhoI/NotI sites of the retroviral vector pMXs-IRES-Neo (Cell Biolabs). Mutations were introduced into the p110 reading frames on these plasmids using the QuikChange site-directed mutagenesis kit (Agilent) and the mutagenesis primers detailed in Supplementary Table 6. All PCR-generated sequences were verified by Sanger sequencing.

**Genetically modified cell lines.** Retrovirus for transduction of BaF3 or Ramos cells was generated by liposomal transfection of pMXs-IRES-Neo expression constructs into Phoenix-Ampho cells that express retrovirus replication and amphotropic envelope proteins. For transfection, 20 μg of plasmid DNA and 60 μl of lipofectamine 3000 (Thermo Fisher) in 1.5 ml of OptiMEM (Gibco) were added to ~70 % confluent Phoenix-Ampho cells in a 10 cm cell culture dish. Following the addition of OptiMEM for 6 h and incubation in DMEM, virus-containing supernatants were collected after 72 h and used directly for infection or stored at −80 °C. Retroviral supernatants with 2 μg/ml polybrene were added to $2 \times 10^5$ BaF3 or Ramos cells on six-well plates for centrifugation at 800×g for 2 h. After cultivation in RPMI-1640 for 2 days, BaF3 or Ramos cells were selected with 1.5 μg/ml geneticin for 6–10 days. To verify the integration of p110δ cDNA in the genome of transduced BaF3 or Ramos cells, templates for sequence verification were amplified from isolated genomic DNA using primers with specificity for the transduced cDNA (Supplementary Table 6). In addition, we demonstrated overexpression of p110δ and expression of HA tag in Western blots.

**Cell viability assay.** Where applicable, transduced BaF3 cells selected for vector-encoded geneticin resistance were additionally deprived of IL-3 for several days prior to viability assessment, as commonly performed in the analysis of oncogenic kinases in this system[48]. Subsequently, they were seeded on 96 wells at a density of $10^5$ cells per well and grown in medium without IL-3 for 96 h. As a surrogate of viable cell numbers, ATP levels were determined using the CellTiter-Glo (CTG) assay (Promega) with luminescence measurements on a FLUOstar Optima plate reader (BMG Labtech) or metabolic activity was measured using the XTT assay (Roche).

**Anchorage-independent growth of fibroblasts.** Colony formation in soft agar by NIH3T3 cells stably expressing p110 isoform variants was examined. Isogenic NIH3T3 cells were trypsinized and suspended at a density of $2.5 \times 10^4$ cells per six-well in a medium containing 10% FBS and 0.5% agarose and plated onto a bottom layer containing 0.6% agarose. After 4 weeks of incubation, three images per well were taken at 25-fold magnification and analyzed using Image J software to determine number of colonies with areas of at least 1200 μm².

**Drug library**. PI3Ki and control substances were purchased as a custom library from Sellekchem (Munich, Germany). Leniolisib was kindly provided by Dr. Christoph Burkhart, Novartis Pharma AG, Switzerland. The compounds were solved in DMSO and stored frozen as 10 mM stocks. The concentration of DMSO as a solvent for inhibitors was kept below 0.1% in cell cultures.

**Drug sensitivity assessment**. For drug library assessment, cells were treated with drugs in seven-point, fourfold serial dilutions ranging from 10 μM down to 2 nM that were prepared with a Felix pipetting station (Analytik Jena). With the same device, compound dilutions were transferred to white 384-well cell culture plates in a volume of 5 μl per well for four replicate measurements. Treatment with 10% DMSO was used for no-growth controls. Isogenic BaF3 cells selected for vector-encoded geneticin resistance were incubated in a medium without IL-3 for several days prior to addition to the 384-well plates with pre-dispensed inhibitors using a MultiFlo FX liquid dispenser (BioTek). Cells were seeded at a density of $10^4$ cells per well (50 μl) and incubated for 72 h. Subsequently, cell viability was determined using the CTG assay. The obtained data were normalized to untreated controls and analyzed with the R package DR4PL using a four-parameter logistic model for the calculation of $IC_{50}$ values. For the final evaluation, the median of four or three library assessments was used for experiments with cells expressing variants of p110δ or p110α, respectively. With the DLBCL cell lines TMD8 and HBL-1, the drug library was assessed twice for the calculation of mean $IC_{50}$ values.

**Signaling analyses**. Ramos cells were deprived of serum for 2 h and incubated with inhibitors for 20 min prior to the addition of 10 μg/ml of goat anti-human IgM F(ab')$_2$ for another 10 min. The resulting Ramos cells with activated BCR were used for signaling analyses by immune-blotting or phospho-specific flow cytometry or cytokine content of culture supernatants. The same treatment scheme was used for BaF3 cells, however using 2 h incubation with 10 ng/μl IL-3 after serum deprivation.

Detection of signaling molecules in Western blots was performed using the Odyssey imager (Licor, Bad Homburg, Germany). Cell lysates containing 30 μg of total protein were immunoblotted and probed by specific primary and fluorescence-labeled secondary antibodies (Supplementary Table 7 and Supplementary Fig. 12).

**Cytokine quantification**. Levels of the chemokines CCL3, CCL4, and TNF-α were determined in culture supernatants after stimulation of $1 \times 10^6$ Ramos cells/ml with anti-IgM (2 μg/mL) for 24 h by means of ELISA kits (Affymetrix, eBioscience) by means of absorbance readings at 450 nm on a FluoSTAR Optima Plate Reader. Standard curves covered concentration ranges from 16 to 2000 pg CCL3 or CCL4/ml or 15.6 to 1000 pg TNFα/ml.

**Chemotaxis assay**. Transwell-migration towards 100 ng/ml of the chemokine CXCL12 was determined essentially as previously described in ref. [42]. Briefly, $2 \times 10^5$ isogenic BaF3 cells suspended in 100 μl of medium were added to transwell culture inserts with a diameter of 6.5 mm and a pore size of 5 μm (Corning, Amsterdam, The Netherlands), that were transferred to the bottom wells containing medium with or without CXCL12 (Peprotech, Rocky Hill, NJ, USA). After 4 h of incubation, cell densities in the lower chambers were counted using a MACSquant flow cytometer (Miltenyi, Bergisch Gladbach, Germany).

**Molecular modeling**. Human PI3Kδ crystal structure was used as a template (PDB entry 6PYR) to build a homology model formed by the helical and kinase domain using the SWISS-MODEL web-based server[49]. The model was energy-minimized by a steepest-descent algorithm in the GROMACS program 2020.4[50]. The root mean square deviation (RMSD) between the model and template is less than 0.3 Å and the quality of the model was validated with PROCHECK[51]. A single-point mutation (methionine) at residue I777 was introduced using the SCWRL4 program with default parameters[52]. As an indication of the high quality of both theoretical models, overall quality factors of 94.43 and 96.74 were obtained for p110δ-wt and p110δ-I777M models, respectively. These two structures were considered as the input structures to perform MD simulations in the next step. These two apo models (wt and mutant) were used to superimpose the coordinates of three small molecule inhibitors from the following PDB entries: idelalisib (4EX0), copanlisib (5G2N), and ZSTK474 (2WXL). A subsequent energy minimization was again performed for each system.

**Quantum mechanical calculations**. Geometry optimization, frequency calculations, and population analyses of the three small molecule inhibitors were performed with the Gaussian 16 package of programs[53] using the B3LYP functional and the 6-31 G(d) basis set. Geometry optimization and frequency calculations were carried out in solution, using the SMD continuum model and water as solvent. SMD is considered a universal solvation model, due to its applicability to any charged or uncharged solute in any solvent or liquid medium[54]. Vibrational analysis indicates that geometries correspond to minima. Computed electrostatic potential (ESP) derived atomic charges were used later for MD simulations.

**Classical MD simulations**. A summary of the simulations is presented in Supplementary Table 5. A total of 8 μs simulations were performed using the all-atom additive CHARMM36 protein forcefield[55] and TIP3 water model[56] with GROMACS 2020.04. Initial complexes were solvated in a rhombic dodecahedron box, with a minimum distance of 10 Å between the structure and the box boundaries. Sodium and chloride ions were added to neutralize the system to an ionic strength of 0.15 mol/L. The systems were equilibrated for 5 ns in the NVT ensemble with restrained heavy atoms, and for 5 ns in the NPT ensemble without restraints. The temperature was stabilized at 310 K using a V-rescale thermostat[57] and pressure at 1 atm by Parrinello-Rahman barostat[58], respectively. Production MD simulations were run on a GPU (GeForce RTX 3090, Cuda 11.4), with a time step of 2 fs. Electrostatic interactions were calculated using the Particle Mesh Ewald method[59] and LINCS algorithm[60] was used for bond constraints. Input and output coordinate files of production simulations are presented in pdb format (Supplementary Data 1).

**Trajectory analyses**. Convergence analysis of equilibrated and production simulations were performed in GROMACS (Supplementary Fig. 13a–d). Representative structures were extracted from trajectories using gmx cluster tool in GROMACS with a single linkage algorithm applying different cut-offs. The structures chosen were the center of the clusters (the structure with the smallest average root mean square deviation from all other structures of the cluster). All representative structures are presented in Supplementary Data 1. Distances and angles were calculated using GROMACS tools gmx distance and gmx angle and processed and analyzed in R[61]. Pymol was used for visualization and figure preparation[62] and PyVol[63] was used to calculate binding pocket volumes.

**Statistics and reproducibility**. We have shown the individual measurements from replicates on the graphs. Mean values and standard deviations were calculated, as well as p values in two-sided unpaired T-tests. Numerical values used for the calculations are provided in the Supplementary Data 2 file.

Wilcoxon signed-rank test with a statistical significance of 0.05 was used to account for significant differences between wt and I777M systems in molecular dynamics simulations. Although we only performed a single MD simulation per condition, our simulations were able to cover a considerable timescale of microseconds. Additional replicas would be needed to ensure reproducibility.

**Reporting summary**. Further information on research design is available in the Nature Portfolio Reporting Summary linked to this article.

## Data availability

Source data for Figures can be found in Supplementary Data 2. All data supporting the findings of this study are available within the paper and its Supplementary Information.

## Code availability

MD simulation input and output files can be found in Supplementary Data 1. The code used in the current study is mentioned in the Methods.

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

## Acknowledgements

This study was supported by grants from the Deutsche José Carreras Leukämie-Stiftung (DJCLS R19/2016 and DJCLS 15R/2019) and the Köln Fortune program of the University of Cologne (379/2021) to G.K. and from the Exzellenz initiieren (EI)—Stiftung Kölner Krebsforschung to R.R.-R. We acknowledge support for the Article Processing Charge from the DFG (German Research Foundation, 491454339).

## Author contributions

G.K., R.R.-R., M.H., F.H., S.J.B., and C.P.P. designed and supervised the study; F.H., L.N., and F.L. performed experiments; F.H., R.R.-R., M.F.-M., and M.R. carried out computational work and MD simulations; J.R.A.-I. performed quantum mechanical

calculations; F.H., G.K., R.R.-R., and M.H. analyzed and interpreted data; G.K., F.H., and R.R.-R wrote the paper. All authors read, revised, and approved the paper.

## Funding

## Competing interests
This study was partially supported with research funding from Gilead Sciences to M.H. The remaining authors declare no competing interests.
