## [Peer Review File · Communications Biology]

Reviewers' comments:

Reviewer #1 (Remarks to the Author):

This manuscript reports on the development of cell based systems for dissection of PI3K inhibitor-related activities using isogenic cell lines that impart modified sensitivity to PI3K activity with respect to various functions, and modified sensitivity or resistance to PI3K inhibitors. The data is wholly original and both the methods and the results will be of use to researchers in the field and more broadly.

My two main suggestions are that the title needs to be changed and the manuscript should be more clearly edited to better reflect the actual content. The title specifically is basically misleading as to what the manuscript is about. I misinterpreted it describing an observation of a clinical relevant mutation that renders inhibitors inactive.

In the manuscript proper, the introduction is very brief and doesn't really set up the motivation for the work that follows. It actually includes a summary of the results mixed in with the background. This really should be re-drafted with more substantive background and demarcation of the study being introduced.

The results section itself is very interesting and a valuable resource. It is very pleasing to see the mix of mutagenesis studies and compound library screening in the one study. The Figures that plot these results (Fig 2) are very complex and I wonder if there is a simpler portrayal, but I couldn't think of it. It feels like they need an explanatory note for each compound. The coloured compound names need explanation in the Figure legend. In some ways the system created is a bit after the event, as PI3Kdelta inhibitors are well established but there is ongoing interest in next-generation ligands such as dual inhibitors. The authors could indicate more strongly where their systems will have utility in the future.

The molecular dynamic study is very interesting and drills to the simple concept that propeller-inhibitors will be influenced by the ability to form the cryptic pocket formed by Met and Trp residues, but also that non-propeller compounds might access other non-apo conformers. As a more extended analysis the authors could connect more to other like-analyses of PI3K isoform by mutagenesis such as where the formation of the cryptic pocket is also dictated by the residue adjacent the methionine - Phe in PI3Kdelta (Zheng et al ACS Chem. Biol. 2013, 8, 679–683). Also, relative the binding of ATP under the mutation would seem to have potential to up-regulate kinase activity by the I771M mutation and this could have been modelled.

Reviewer #2 (Remarks to the Author):

Hassenrück and colleagues have developed a high-throughput cell-based assay system for the assessment of PI3K inhibitor isoform selectivity and drug resistance. They use this system to assess a panel of PI3K inhibitors with varying isoform selectivity profiles, explore the role of PI3K delta in BCR-signalling and chemokine secretion, and the functional consequences of a resistance mutation, I777M. While presenting some interesting results, the authors choose to focus predominantly on understanding changes in inhibitor binding to the resistance mutant, I777M. My major criticism, however, is that this mutant is not clinically relevant and indeed, resistance to idelalisib has been found to be mostly based on compensatory mechanisms and other pathway changes, rather than

PI3Kd mutations. The relevance for this part of the work, therefore, is limited. The manuscript would benefit from a significant rewrite and restructure to present the work as the development of a cell-based assay to investigate isoform selectivity and/or resistance mutations if/when they arise, using I777M as a test case to show feasibility or focusing on the insights gained on PI3Kd's role in BCR-signalling and chemokine secretion.

General comments:

A paragraph in the introduction and/or discussion summarising other systems that exist for measuring cellular IC50s of PI3Kd inhibitors and highlighting how they differ from the one employed here.

The differential regulation of chemokine secretion by PI3Kdelta overexpression is an interesting finding. It is also particularly interesting that idelalisib seems to block all the expression of all three chemokines. This is also worth a paragraph in the discussion.

Specific points:

Line 52-54: This paragraph is rather misleading as it presents the rationale for the current study as being "acquired resistance" to idelalisib, when the mutant under investigation has not been shown to be involved in resistance.

Line 106: The I777M mutation has a significantly higher transformation capacity than WT or E1021K. Can the authors propose a mechanism for this? Do the molecular dynamics studies shed any light on why the I777M may be more active?

Are the expression levels of the PI3Ka and PI3Kd variants equivalent in each of the transduced cell lines? A western blot showing PI3K expression levels should be done to fully characterise the cell lines.

Fig 1A: The specific statistics represented by the box plot should be expanded in the figure legend. Are the p-values for PI3Kd-wt and PI3Kd-E1021K+I777M correct? The values look like they may have been swapped inadvertently.

Fig. 1A and Ext. Fig. 1B: can the authors clarify the difference between these two results? The main text suggests Fig. 1A is the result of geneticin selection, and Ext. Fig. 1B is the result of geneticin & IL3-independent viability, but the figure legends for Fig. 1A only mention IL3-deprivation and Ext. Fig. 1B mentions antibiotic & IL3-deprivation. The results for whichever selection method was used moving forward should be in the main figure.

If the rare mutation E1021K shows only a marginal improvement compared to wild-type, would it not make more sense to characterise the inhibitors on a PI3Kd wild-type background, given this is the most common clinical setting?

Line 106-111: how does the anchorage-independent growth assay "clarify the pronounced increase in IL-3-independent viability owing to the binding pocket mutation"? Was the purpose to confirm it was a mutation-effect and not simply a system artefact?

Fig. 1B&C: Is %viability calculated in the same way as for Fig. 1A or are the transduced cells normalised to a no compound control? This should be clarified, and if not normalised to the no compound control, then the %viability for this control should be included.

Is the idelalisib induced cell death observed in the parental cells PI3Kdelta dependent or non-specific toxicity?

The effects of the other inhibitors on the parental cell line with IL3 should also be shown.

Fig. 2: How were cellular IC50 values calculated? Is Fig. 2A based on cells with PI3Kd E1021K and

PI3Ka H1047R or WT? This should be specified. Are these mean or median values from how many replicates?

Fig. 2B: Has the PI3K isoform expression been characterised in these lines? A western blot or some other form of quantification of the expression of the different isoforms should be done for these cell lines.

Supp Table 2&3: Why are median IC50 values quoted rather than mean?

Supplementary figures: all but Supp Fig 1 are incorrectly referred to in the text. Please fix.

Line 177: The significance of mutant I825V is not mentioned anywhere in the text. The rationale behind this mutant should be added.

Fig. 3A: Vector control should be shown on these graphs, not only in supplementary

Ext. Fig. 4A: Despite negligible amounts of PI3Kd in the vector control, idelalisib reduces pAKT levels. Is this all due to endogenous PI3Kdelta? Are there other PI3K isoforms present?

Line 206-207: It is interesting that overexpression of PI3Kd increases only CCL3, but not CCL4 or TNFa, but treatment with idelalisib reduces expression of all three in a dose dependent manner. Is this due to off-target effects of idelalisib or dependence on other PI3K isoforms? The effect of inhibitors with differing isoform selectivity on the varying chemokines would be worth investigating further.

Line 210: Sup. Figure 4B, which I assume is being referred to here measures metabolic activity rather than cell viability.

Sup. Fig. 4C and Fig3D – these should be combined as one panel, there seems to be no reason to split the data across two figures. The p-values for WT +/- idelalisib pair and I777M +/- idelalisib should be quoted. Comparing the WT and I777M with idelalisib doesn't make sense as the starting points (WT or I777M without idelalisib) are not the same.

Ext. Data Fig. 6A: I think the mutant labels must be incorrect. Presumably the single mutant is H1047R and the double mutant is H1047R/I800M?

Line 239-242. It would be interesting to include one of the inhibitors that showed increased activity against the I777M mutant in the MD studies.

Supp. Table 5: what is the reason for the longer simulation time for I777M apo?

Line 248: Does a comparison between the wt and I777M apo structures yield any insights into why the I777M mutant is more strongly transforming/more active than the wild-type?

Structure figures: The hydrogens should be removed from the structure figures. Where there are hydrogen bonds discussed in the text, an attempt should be made to show these in the figures.

Line 267: Should read Fig. 5D, E

Fig. 6C: This figure is extremely confusing and difficult to interpret. The same colours have been used in one part of the figure to distinguish different inhibitors, and then in the other part of the figure to distinguish different parts of the binding pocket. Consider using one colour scheme for the binding pocket and a different colour scheme for the different inhibitors. A better description for exactly what

is being depicted in the bottom half of the figure is required.

Ext. Data Fig. 7C: "close proximity" should be precisely defined.

Line 295: The results from the copanlisib simulation should be shown in a more detailed figure, potentially in the supplementary.

Line 321: Can the authors expand on what is meant by p110delta selective PTEN activation? Ext. Data Fig. 5D – where this refers to PI3Kdelta selective, the isoform should be indicated in the diagram. It is unclear how this describes a PI3Kd selective, negative feedback loop.

Author rebuttal of reviewer comments

Reviewer #1

This manuscript reports on the development of cell based systems for dissection of PI3K inhibitor-related activities using isogenic cell lines that impart modified sensitivity to PI3K activity with respect to various functions, and modified sensitivity or resistance to PI3K inhibitors. The data is wholly original and both the methods and the results will be of use to researchers in the field and more broadly.

My two main suggestions are that the title needs to be changed and the manuscript should be more clearly edited to better reflect the actual content. The title specifically is basically misleading as to what the manuscript is about. I misinterpreted it describing an observation of a clinical relevant mutation that renders inhibitors inactive.

Thank you for the thoughtful analysis of our manuscript and the constructive suggestions. To improve the manuscript, we have altered the title to be more in keeping with the core findings of the paper: "Functional impact and molecular binding modes of drugs that target the PI3K isoform p110 δ ". In addition, we abandoned the concept of I777M as a resistance mutation in favour of its application as a chemical genetic tool for cell function analysis and an inducer of altered binding pocket conformers.

In the manuscript proper, the introduction is very brief and doesn't really set up the motivation for the work that follows. It actually includes a summary of the results mixed in with the background. This really should be re-drafted with more substantive background and demarcation of the study being introduced.

We agree with the reviewer that the introduction needs to be more substantive. Therefore, we have redrafted the introduction, disentangling background and summary, to provide a more defined context to the underlying biology and molecular pharmacology of the paper and why what we observe is providing additional knowledge to the field.

The results section itself is very interesting and a valuable resource. It is very pleasing to see the mix of mutagenesis studies and compound library screening in the one study. The Figures that plot these results (Fig 2) are very complex and I wonder if there is a simpler portrayal, but I couldn't think of it. It feels like they need an explanatory note for each compound. The colored compound names need explanation in the Figure legend. In some ways the system created is a bit after the event, as PI3K δ inhibitors are well established but there is ongoing interest in next-generation ligands such as dual inhibitors. The authors could indicate more strongly where their systems will have utility in the future.

We appreciate the positive comments from the reviewer. As we feel that it is important to present the data in its complexity, we only improved the readability of *Figure 2*, and added the requested explanation of the colored compound names in the Figure legend (*lines 794-795*). For a different presentation of the data, we have added *supplementary Figures 2C, D*, in which IC₅₀ values obtained with BaF3 cells expressing p110δ-E1021K and the DLBCL cell line TMD8 are compared. Moreover, we have added further remarks on the future utility of the described assay systems in the Discussion (e.g. *lines 328-331, 398-400, 422-424*).

The molecular dynamic study is very interesting and drills to the simple concept that propeller-inhibitors will be influenced by the ability to form the cryptic pocket formed by Met and Trp residues, but also that non-propeller compounds might access other non-apo conformers. As a more extended analysis the authors could connect more to other like-analyses of PI3K isoform by mutagenesis such as where the formation of the cryptic pocket is also dictated by the residue adjacent the methionine - Phe in PI3Kdelta (Zheng et al ACS Chem. Biol. 2013, 8, 679–683). Also, relative the binding of ATP under the mutation would seem to have potential to up-regulate kinase activity by the I771M mutation and this could have been modelled.

We are grateful to the reviewer for the suggestion to consider non-conserved binding pocket residues in addition to the conserved Ile777 in p110δ, since this gives us the opportunity to discuss the molecular basis for isoform selectivity of PI3K inhibitors on a broader scale. For this purpose, we have looked specifically at the non-conserved amino acid F751 in our molecular dynamics simulations of the wild type and I777M mutant of PI3Kδ, particularly regarding the specificity pocket induced by idelalisib. In both simulations, the side chain of residue F751 points away from the active site and does not interfere with idelalisib binding. F751 shows, however, 513 versus 18 out of 1000 frames in wild type versus I777M trajectories within 3Å distance to bound idelalisib reflects, reflecting the impact of the I777M mutation on conformational changes affecting residues quite distant from the mutation site including the neighboring M752. This sharpened our discussion of the remote effect of the I777M mutation on the position of M752 with importance for idelalisib binding (*lines 405-415*).

Following the reviewer's suggestion, we have performed molecular dynamics simulations of ATP binding to wild type and mutant p110δ to obtain clues to the enhanced activity of p110δ-I777M. These showed differences between mutant and wild type in binding poses for ATP as well as total interaction energy, but could not fully explain the increased activity of the mutant. In an appendix to this rebuttal, we provided a Figure for the reviewer in support of the described effects of the mutation, although we have decided to not include these results in the revised manuscript.

Reviewer #2:

We took the liberty to insert numbering of the General Comments and Specific Points for easier cross-reference. Line numbers in the rebuttal refer to the revised version.

Hassenrück and colleagues have developed a high-throughput cell-based assay system for the assessment of PI3K inhibitor isoform selectivity and drug resistance. They use this system to assess a panel of PI3K inhibitors with varying isoform selectivity profiles, explore the role of PI3K delta in BCR-signalling and chemokine secretion, and the functional consequences of a resistance mutation, I777M.

While presenting some interesting results, the authors choose to focus predominantly on understanding changes in inhibitor binding to the resistance mutant, I777M. My major criticism, however, is that this mutant is not clinically relevant and indeed, resistance to idelalisib has been found to be mostly based on compensatory mechanisms and other pathway changes, rather than PI3K δ mutations. The relevance for this part of the work, therefore, is limited. The manuscript would benefit from a significant rewrite and restructure to present the work as the development of a cell-based assay to investigate isoform selectivity and/or resistance mutations if/when they arise, using I777M as a test case to show feasibility or focusing on the insights gained on PI3K δ 's role in BCR-signalling and chemokine secretion.

Thank you so much for the detailed review and numerous useful hints for improving our manuscript. Since the I777M mutation is not relevant for the clinical occurrence of PI3Ki resistance, we have refocused our manuscript to emphasize its role as a tool for dissecting PI3K δ functions and as a means to test structure-guided hypotheses. For this purpose, we have provided a new title and introduction better representing the content of the paper.

General comments:

(1) A paragraph in the introduction and/or discussion summarising other systems that exist for measuring cellular IC50s of PI3K δ inhibitors and highlighting how they differ from the one employed here.

Thank you for your comment, which reminded us of the central importance of this piece of background information for one of the main topics of our manuscript. We have reinforced and expanded the corresponding passages in the discussion (*lines 318-329*), keeping the two existing references (*no. 27 and 28*) that mention the described cellular mechanistic models in the characterization of specific PI3K inhibitors.

(2) The differential regulation of chemokine secretion by PI3K δ overexpression is an interesting finding. It is also particularly interesting that idelalisib seems to block all the expression of all three chemokines. This is also worth a paragraph in the discussion.

Thank you for this positive comment. We agree that both, the strict dependence of cytokine secretion on the activation of B cell receptor signalling as well as its modulation by the engineered p110 δ overexpression that the reviewer noted needed additional explanation and emphasis. Therefore, we have more thoroughly addressed the interesting concept of idelalisib blocking production of these cytokines in a separate paragraph of the discussion (*lines 385-401*).

Specific points:

(1) Line 52-54: This paragraph is rather misleading as it presents the rationale for the current study as being “acquired resistance” to idelalisib, when the mutant under investigation has not been shown to be involved in resistance.

We fully agree with the comment from the reviewer as to the inappropriateness of “acquired resistance” as the rationale of the study. Therefore, we changed the manuscript title and aims and followed the reviewers request to modify the first paragraph of the introduction, to more clearly express the partially unexploited potential of PI3K inhibitors in the treatment of B cell malignancies. However, resistance as such is still mentioned as one of the major limitations of idelalisib treatment.

(2) Line 106: The I777M mutation has a significantly higher transformation capacity than WT or E1021K. Can the authors propose a mechanism for this? Do the molecular dynamics studies shed any light on why the I777M may be more active?

The transformation capacities of the I777M and E1021K mutation were not directly compared, since, at the onset of our study, E1021K was not yet part of the investigated mutation panel. The increased oncogenic potential of combined E1021K and I777M may be partly due to mutual enhancement of the two mutations. We have addressed the transformation capacity of the two gain of function mutations E1021K and I777M at greater detail in our replies to specific comments no. 6 and 7, respectively. To explore the mechanism leading to higher enzyme activity we followed the strategy outlined in our reply to specific point 21.

(3) Are the expression levels of the PI3Ka and PI3Kd variants equivalent in each of the transduced cell lines? A western blot showing PI3K expression levels should be done to fully characterise the cell lines.

We particularly appreciate this comment from the reviewer, since it refers to an essential control of the major cell line model used in our study that had been missing. Therefore, we have provided an additional *External Data Fig. 1A* showing the p110 δ and p110 α expression for the complete panel of isogenic BaF3 cells (*lines 102-104; 969-971*). These Western blots in the first place serve the purpose to ascertain approximately equivalent overexpression of

p110 δ and p110 α variants. In addition, they show that overexpression of one isoform leads to down-regulation of the other. This is a modulation of the expression pattern in the parental cell line, but a reinforcement of the engineered isoform-dependence.

(4) Fig 1A: The specific statistics represented by the box plot should be expanded in the figure legend.

Are the p-values for PI3Kd-wt and PI3Kd-E1021K+I777M correct? The values look like they may have been swapped inadvertently.

Thanks for checking the statistics so diligently. In fact, wrongly assigned p-values had been indicated, namely those referring to values obtained by normalization to parental BaF3 cells. This error has been corrected and the legend expanded as requested (*lines 969-971*). To comply with the editorial guidelines, we now have shown individual values instead of box plots in this Figure with n=6. The illustration is presented as *External Data Fig. 1B* in the revised manuscript, as outlined in comment no. 5.

(5) Fig. 1A and Ext. Fig. 1B: can the authors clarify the difference between these two results? The main text suggests Fig. 1A is the result of geneticin selection, and Ext. Fig. 1B is the result of geneticin & IL3-independent viability, but the figure legends for Fig. 1A only mention IL3-deprivation and Ext. Fig. 1B mentions antibiotic & IL3-deprivation. The results for whichever selection method was used moving forward should be in the main figure.

We are grateful to the reviewer for raising this important point. This comment alerted us of the potentially misleading method descriptions in the previous version that we have corrected (*lines 742-744* and *lines 971-974*). We now refer to preceding versus concomitant IL-3 deprivation, instead of single and so-called double selection, which consists of geneticin selection followed by IL-3 deprivation, because the ensuing test for IL-3 independence also involves IL-3 deprivation. Following the reviewer's advice, we have exchanged the former *Fig. 1A* and the new *Extended Data Fig. 1B*, according to the use of concomitant versus preceding IL-3 withdrawal. The ultimately applied procedure, results in considerably higher percentages of IL-3-independent cell viability than concomitant IL-3-withdrawal.

(6) If the rare mutation E1021K shows only a marginal improvement compared to wild-type, would it not make more sense to characterise the inhibitors on a PI3Kd wild-type background, given this is the most common clinical setting?

This is a good question from the reviewer that will help to overcome weaknesses in the previous presentation of the results. For this purpose, we have performed the Figure rearrangement explained in specific point no. 5. In the new *Extended Data Fig. 1A*, for which concomitant IL-3 withdrawal was used, the signal increase by the E1021K mutation admittedly appears small. This may be caused by incomplete selection as well as continued

background growth. However, this increase is essential for robust IL-3-independent growth and survival under the finally used conditions. In turn, prolonged IL-3 deprivation of BaF3 cells can result in unspecific IL-3-independence leading to a loss of idelalisib sensitivity. This was occasionally observed with BaF3 cells expressing p110 δ -wt, but never with p110 δ -E1021K. Thus, for p110 δ as well as for p110 α -wt, stable IL-3-independent viability of isogenic BaF3 cells can only reproducibly be achieved with additional C-terminal mutations. For a better description of this situation, we have modified the text in the first paragraph of the results section (*lines 104-106 and 117-118*). Moreover, we feel that using the less common E1021K mutant is justified, since we are using cellular model systems and trying to unpick the biology and not specifically in relation to the clinic.

(7) Line 106-111: how does the anchorage-independent growth assay “clarify the pronounced increase in IL-3-independent viability owing to the binding pocket mutation”? Was the purpose to confirm it was a mutation-effect and not simply a system artefact?

As pointed out in specific point no. 6, E1021K is a prerequisite for the described isoform-selective assay system, whereas the oncogenic properties of I777M were an unexpected observation in BaF3 cells that we simply wanted to evaluate also in a different oncogenicity assay. Therefore, we have corrected the misleading wording of the previous manuscript version (*line 112*). In both investigated systems, the observed gain of function is clearly linked to the I777M substitution and not a system artefact.

(8) Fig. 1B&C: Is %viability calculated in the same way as for Fig. 1A or are the transduced cells normalised to a no compound control? This should be clarified, and if not normalised to the no compound control, then the %viability for this control should be included. Is the idelalisib induced cell death observed in the parental cells PI3Kdelta dependent or non-specific toxicity?

The effects of the other inhibitors on the parental cell line with IL3 should also be shown.

Thank you for the reviewer’s comment. Yes, the viability was calculated in the same way for the entire *Fig. 1* of the revised version, and the concentration response diagrams in *Fig. 1C* were obtained by additional normalization to untreated samples (*lines 744-745*). As to the untreated samples in the new *Extended Data Fig. 1B*, in fact, a different normalization is used, namely to the respective isogenic instead of parental BaF3 cells (*lines 973-974*).

Regarding the effects on parental BaF3 cells with IL-3, we did not observe relevant cell death due to non-specific toxicity throughout the investigated range of PI3Ki. Therefore, we have included the results with parental BaF3 cells for idelalisib to illustrate this fact (*Fig. 1B*) and for the control substances with $IC_{50} < 10 \mu\text{M}$ shown in *supplementary Figure 6*, but not systematically for all investigated PI3Ki.

(9) Fig. 2: How were cellular IC50 values calculated? Is Fig. 2A based on cells with PI3Kd E1021K and PI3Ka H1047R or WT? This should be specified. Are these mean or median values from how many replicates?

Fig. 2B: Has the PI3K isoform expression been characterised in these lines? A western blot or some other form of quantification of the expression of the different isoforms should be done for these cell lines.

We thank the reviewer for this clarification. We have included in the methods the way we calculated the IC₅₀ values (*lines 487-492*). As pointed out in specific point no. 6, robust IL-3 independence of BaF3 cells, as used in *Figure 2*, only works with the C-terminal mutations. This was more clearly indicated in the Figure legend (*line 787*). As for *Fig. 2B*, we have conducted Western Blotting of p110 δ and p110 α expression in the whole set of isogenic BaF3 cells, which is in *Extended Data Figure 1A*, as mentioned in a previous rebuttal answer (specific point no. 3).

(10) Supp Table 2&3: Why are median IC50 values quoted rather than mean?

We have reported median IC₅₀ values of four or three biological replicates for BaF3 cells expressing p110 δ -E1012K or p110 α -H1047K, respectively, while presentation of data from just one screening run is not uncommon in such inhibitor profiling efforts. The choice between using the mean or median as a measure of central tendency depends on the specific characteristics of the data. Since the mean is sensitive to outliers or extreme values, and can be greatly affected by them, we chose the median because it represents more robust statistics in this case. The error indicated refers to the median absolute deviation.

(11) Supplementary figures: all but Supp Fig 1 are incorrectly referred to in the text. Please fix.

Thank you for noticing this error, it has been corrected.

(12) Line 177: The significance of mutant I825V is not mentioned anywhere in the text. The rationale behind this mutant should be added.

We are grateful to the reviewer for bringing up this point. In fact, the significance of the gatekeeper mutation I825V had been mentioned only in the legends of *Extended Data Figs. 3 and 4* and in the Discussion of the previous manuscript version. A short explanation of the I825V mutant has now been added to the introduction (*lines 82-84*).

(13) Fig. 3A: Vector control should be shown on these graphs, not only in supplementary.

Due to limited sample slots, the Western Blots with serially diluted idelalisib were performed on separate gels containing samples of only two of the four investigated isogenic cell lines. Therefore, the quantified signals can only be directly compared among the cell types on the same gel.

(14) Ext. Fig. 4A: Despite negligible amounts of PI3K δ in the vector control, idelalisib reduces pAKT levels. Is this all due to endogenous PI3K δ ? Are there other PI3K isoforms present?

Although our mechanistic cell line model relies on IL-3-independent growth and survival of engineered isogenic cells overexpressing p110 isoforms, the endogenous p110 δ expression of parental BaF3 and Ramos cells is in fact sufficient for significant reduction of AKT phosphorylation by idelalisib, while even NIH3T3 cells with apparently absent p110 δ expression show weak reduction of pAKT by idelalisib (*External Data Fig. 5*). Basal and engineered expression of p110 δ and p110 α in isogenic BaF3 cells is shown in the new *Extended Data Figure 1A*.

(15) Line 206-207: It is interesting that overexpression of PI3K δ increases only CCL3, but not CCL4 or TNF α , but treatment with idelalisib reduces expression of all three in a dose dependent manner. Is this due to off-target effects of idelalisib or dependence on other PI3K isoforms? The effect of inhibitors with differing isoform selectivity on the varying chemokines would be worth investigating further.

This is an astute observation by the reviewer. We agree that the context of chemokine assessment is interesting, but we believe it to be out of the remit of this manuscript, especially as far as overexpression effects are concerned. Cytokine secretion in Ramos cells is a consequence of anti-IgM stimulation of B cell receptor signaling, in which p110 δ is the preferentially involved PI3K isoform. As to the explored cell functions in inhibitor profiling, we focused on PI3K isoform-driven growth and survival rather than on cytokine secretion.

(16) Line 210: Sup. Figure 4B, which I assume is being referred to here, measures metabolic activity rather than cell viability.

The reviewer's comment highlights an intricacy that was not properly described in the manuscript. In fact, *suppl. Fig. 4* uses data obtained with the XTT assay of metabolic activity and not the Cell Titer Glo assay based on ATP amounts. However, both assays reflect total viable cell numbers, which result from cell growth and survival during the incubation. We have adapted the results and methods sections of the manuscript accordingly (*line 208 and lines 463-465*).

(17) Sup. Fig. 4C and Fig3D – these should be combined as one panel, there seems to be no reason to split the data across two figures. The p-values for WT +/- idelalisib pair and I777M +/- idelalisib should be quoted. Comparing the WT and I777M with idelalisib doesn't make sense as the starting points (WT or I777M without idelalisib) are not the same.

Thank you for your thoughts on our migration assay data. We agree that our previous data presentation appeared redundant and therefore have redesigned it. We disagree with the

notion that wt and I777M cannot be compared for lack of an idelalisib-treated no migration control, since spontaneous migration (without chemoattractant) is about 1000-fold lower than migration to CXCL12 and, in fact, too low to reasonably evaluate its further reduction by idelalisib treatment at greater detail. Therefore, we maintain that the differentially reduced migration in the presence of idelalisib in isogenic BaF3 cells is an important consequence of the I777M mutation and have shown the percentages of the migration of idelalisib-treated versus untreated isogenic BaF3 cells on a linear scale (new Fig. 4E). For the sake of completeness, we also show the huge increase of migration owing to CXCL12 as a chemoattractant on a logarithmic scale in a second panel of the main Figure (former suppl. Fig. 4C, new Fig. 4D).

(18) Ext. Data Fig. 6A: I think the mutant labels must be incorrect. Presumably the single mutant is H1047R and the double mutant is H1047R/I800M?

Thank you for noticing this error, it has been corrected.

(19) Line 239-242. It would be interesting to include one of the inhibitors that showed increased activity against the I777M mutant in the MD studies.

Thank you for the suggestion to expand the range of inhibitors for molecular interaction analysis. While selection according to cellular resistance, potency and isoform selectivity yielded three prototypes representing clearly different molecular mechanisms, we think that sensitization owing to the I777M mutation is a less suitable criterion for identifying inhibitors with differing binding modes. Importantly, the apparent sensitization is comparatively modest and likely caused by p110 δ overexpression in the mechanistic cell line model cells, as outlined in the Discussion and in specific point no. 27. Moreover, the eligible inhibitors according to this criterion, i.e. predominantly dual-specific PI3K-mTOR inhibitors, show either low potency or selectivity for p110 δ . Therefore, we concentrated our MD simulation efforts during the revision period on the binding of ATP (in response to specific points no. 2 and 21) rather than on an additional PI3K inhibitor.

(20) Supp. Table 5: what is the reason for the longer simulation time for I777M apo?

Since in the 1 μ s simulation of I777M apo we found two highly populated clusters, we extended the simulation by 1 μ s more to rule out that the sampling of the conformational space was not sufficient and the system was trapped in a local minimum. Cluster analysis of the 2 μ s simulation resulted in a single highly populated cluster, from which a representative structure was extracted, as described in the Methods section.

(21) Line 248: Does a comparison between the wt and I777M apo structures yield any insights into why the I777M mutant is more strongly transforming/more active than the wild-type?

Presumably because our MD simulations of the non-helical parts of the kinase domain cannot adequately capture important mechanisms like access to membrane-resident substrate and interaction with the regulatory subunit, we could not derive any clues to increased enzyme activity from comparisons of mutant and wild type apo structures (*Supplementary video 1*). To better resolve a potential contribution of altered ATP binding, we therefore performed molecular dynamics simulations of ATP bound to p110 δ -wt and p110 δ -I777M during the revision period. Although these simulations showed changes owing to the I777M mutation in the binding poses for ATP as well as in total interaction energy, we feel that these changes cannot fully explain the increased activity of the mutant. Therefore, and since the gain of function owing to the I777M substitution is an incidental side aspect of our study, we have decided to not include these results in the revised manuscript, but also could not fully explain the increased activity of the mutant. Therefore, we have not included these results in the revised manuscript, but in the appendix accompanying this rebuttal.

(22) Structure figures: The hydrogens should be removed from the structure figures. Where there are hydrogen bonds discussed in the text, an attempt should be made to show these in the figures.

Hydrogen atoms are now only shown in the *Extended Figures 7 and 8*. We show the atoms on purpose, since we measured some hydrogen bonds that are discussed in the text. In general terms, we consider that including hydrogen atoms is an advantage provided by molecular dynamics over crystallographic structures where no hydrogens are present.

(23) Line 267: Should read Fig. 5D, E

Thank you for noticing this error, it has been corrected.

(24) Fig. 6C: This figure is extremely confusing and difficult to interpret. The same colours have been used in one part of the figure to distinguish different inhibitors, and then in the other part of the figure to distinguish different parts of the binding pocket. Consider using one colour scheme for the binding pocket and a different colour scheme for the different inhibitors. A better description for exactly what is being in the bottom half of the figure is required.

We thank the reviewer for the advice concerning this Figure, which has helped to make it better understandable. For this purpose, we decided to use only one color scheme, namely that referring to binding pocket parts. For better distinction from other color schemes in the paper we have also changed the color indicating the specificity region to yellow. The description in the Figure legend has been adapted.

(25) Ext. Data Fig. 7C: "close proximity" should be precisely defined.

The precise minimal distance of residues shown is 3 Å.

(26) Line 295: The results from the copanlisib simulation should be shown in a more detailed figure, potentially in the supplementary.

Thank you for the suggestion. In response, we have added a separate paragraph about the MD simulations with copanlisib to the results section (*lines 288-295*). This includes reference to the new *Supplementary Video 5* that shows the undisturbed binding of copanlisib to p110 δ -I777M.

(27) Line 321: Can the authors expand on what is meant by p110delta selective PTEN activation? Ext. Data Fig. 5D – where this refers to PI3Kdelta selective, the isoform should be indicated in the diagram. It is unclear how this describes a PI3Kd selective, negative feedback loop.

Thank you for pointing out unclear information in this discussion point and the related signaling scheme. The confusion may be partly due to an erroneous mix up of references in the previous manuscript version (PMID 19033389 used instead of PMID 17581634). This mistake has been corrected. The proper reference no. 37 contains the genetic and pharmacological demonstration, that CSF-1-stimulated RhoA and PTEN activation in macrophages depend on p110 δ rather than other class I isoforms. To further clarify the negative feedback occurring upon p110 δ overexpression, we have modified *External Data Fig. 5D* and its legend, as well as corresponding text passages in the Discussion (*lines 358-362*).

Appendix

We quantified how much the ATP pose changed over the simulation by computing the root mean square deviation (RMSD) of ATP heavy atoms (Fig. 1A). The ATP position changed drastically in the mutant compared to the wild type with median values of 3 Å and 2 Å, respectively. Moreover, in order to quantify the strength of the interaction between the ATP and p110δ, the total interaction energy of both systems was calculated (Fig. 1B). After propagating the error, the total energy for the wild type is -976.79 ± 2.9 kJ mol⁻¹, while for the I777M mutant it is -946.79 ± 1.7 kJ mol⁻¹. Together, we found that the I777M mutation affects the natural binding of ATP to p110δ.

Fig. 1: Molecular dynamics simulations of ATP and 2 Mg²⁺ ions bound to wild type and mutant p110δ. (A) RMSDs of wild type (grey) and I777M mutant (red) as a function of time. (B) Stacked bars show the total interaction energies of p110δ-ATP from MD simulations. Median values of short-range Coulomb (Coul-SR) and short-range Lennard-Jones (LJ-SR) energies are represented by black and brown color, respectively.

REVIEWERS' COMMENTS:

Reviewer #1 (Remarks to the Author):

The authors have addressed the topics raised in my previous review and the other reviewer thoroughly, which I believe has improved the clarity of the manuscript with respect to its purpose and its findings.

Reviewer #2 (Remarks to the Author):

The authors have addressed all my concerns in detail and I now recommend the manuscript for publication.